# Trial-history biases in evidence accumulation can give rise to apparent lapses in decision-making

Diksha Gupta [1,3] ✉, Brian DePasquale[1,4], Charles D. Kopec[1] & Carlos D. Brody [1,2] ✉

Trial history biases and lapses are two of the most common suboptimalities observed during perceptual decision-making. These suboptimalities are routinely assumed to arise from distinct processes. However, previous work has suggested that they covary in their prevalence and that their proposed neural substrates overlap. Here we demonstrate that during decision-making, history biases and apparent lapses can both arise from a common cognitive process that is optimal under mistaken beliefs that the world is changing i.e. nonstationary. This corresponds to an accumulation-to-bound model with history-dependent updates to the initial state of the accumulator. We test our model's predictions about the relative prevalence of history biases and lapses, and show that they are robustly borne out in two distinct decision-making datasets of male rats, including data from a novel reaction time task. Our model improves the ability to precisely predict decision-making dynamics within and across trials, by positing a process through which agents can generate quasi-stochastic choices.

It has long been known that experienced perceptual decision makers deviate from the predictions of optimal decision-theory, displaying several suboptimalities in their decision-making. Among the most pervasive of these is the dependence of behavior on the recent history of observed stimuli, performed actions, or experienced outcomes, despite it being disadvantageous and leading to worse performance[1–18] (schematized in Fig. 1a top). History biases may arise due to a strategy that is optimized for naturalistic settings, where continual learning of priors, action-values, or other decision variables helps agents adapt to changing environments, but is maladaptive in experimental settings where the statistics of the environment are stationary[19,20]. To date, decision-theoretic models have accommodated history biases by modeling them as a biasing factor on the perceptual evidence that drives choices[3,12,13,21–26]. In the predominant conceptualization of these models, history biases can be overcome with sufficient perceptual evidence.

A second widely-recognized but less studied suboptimality is the tendency to "lapse", or make (asymptotic) errors that are immune to strong evidence[3,4,11,27–33] (schematized in Fig. 1a bottom). Because lapses appear to be evidence-independent, they are assumed to arise from nuisance mechanisms that are separate from the perceptual decision-making process and are often imputed to ad-hoc noise sources such as inattention, motor errors etc.

However, several recent results suggest that these two suboptimalities may be linked in their origin. In primates, learning reduces dependence on recent trial history[2] as well as lapse probabilities[28]. Intriguingly, mice trained on a visual detection task showed higher levels of history dependence on sessions with higher lapse probabilities[3]. Moreover, lapses occur in runs (i.e. display Markov dependencies), rather than occurring with the traditionally assumed independent probabilities across trials[34]. Furthermore, lapses have been proposed to reflect forms of exploration[32] that are sensitive to

[1]Princeton Neuroscience Institute, Princeton University, Princeton, NJ, USA. [2]Howard Hughes Medical Institute, Princeton University, Princeton, NJ, USA. [3]Present address: Sainsbury Wellcome Centre, University College London, London, UK. [4]Present address: Department of Biomedical Engineering, Boston University, Boston, MA, USA. ✉e-mail: diksha.gupta@ucl.ac.uk; brody@princeton.edu

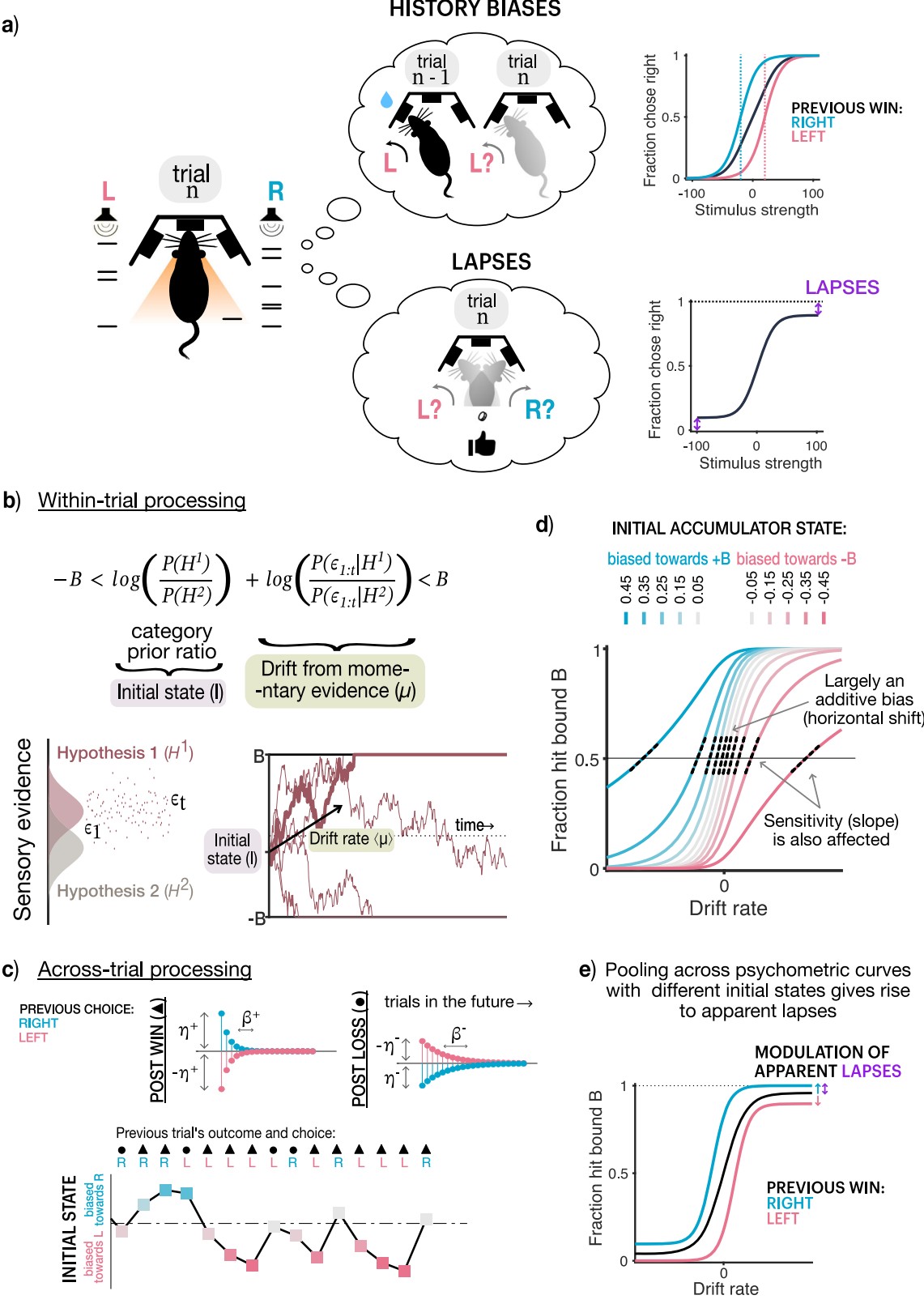

trial-by-trial updates of variables such as action value. Likewise, neural perturbations of secondary motor cortex and striatum in rodents have been shown to substantially impact both lapses[32,35–39] and trial-history influences on decisions[39,40]. Together, these observations challenge the assumption that history biases and lapses have independent causes and raise the possibility that some of the variance ascribed to lapses emerges from history dependence.

In this work, we explore the idea that history biases reflect a misbelief about non-stationarity in the world, and demonstrate that normative decision-making under such beliefs gives rise to choices that are both history-dependent and appear to be evidence-independent (i.e. akin to lapses). This corresponds to an accumulation to bound process with a history dependent initial state. We fit this model to a large dataset of choices made by 152 rats trained on an

**Fig. 1 | Trial history-dependent initial states give rise to apparent lapses.**
**a** Schematic of two common suboptimalities: history biases (top) and lapses (bottom). (Left): Rat making one of two decisions (left, right) based on accumulated sensory evidence (clicks on either side). (Top left): History biases i.e. an inappropriate influence of the previous trial (n-1) on the current decision (n) in addition to sensory evidence. (Top right): Typically assumed effect of history bias on the psychometric curve, shifting it horizontally around the inflection point. (Bottom left): Lapses i.e. a tendency to make seemingly random choices irrespective of sensory evidence. (Bottom right): Typically assumed effect of lapses on the psychometric curve, vertically scaling its asymptotes (Figure adapted with permission from Bingni W. Brunton et al., Rats and Humans Can Optimally Accumulate Evidence for Decision Making. Science 340,95-98(2013). DOI:10.1126/science.1233912) **b** Normative model of within-trial processing. (Top) Optimal decision rule that chooses when the summed log-ratios of priors and likelihoods exceeds one of two decision bounds, corresponding to a drift-diffusion process. (Bottom left): Generative model, where one of two hypotheses (H1, H2) produce noisy evidence over time ($\epsilon_t$). (Bottom right): A sample trajectory based on noisy evidence (bold line), and alternate trajectories (thin lines) based on noisy instantiations of the same drift rate (black arrow). **c** Model of across-trial processing that accommodates prior updates. Past choices and outcomes can affect the initial state with different magnitudes ($\eta$) and timescales ($\beta$) depending on whether they were wins/losses (top left/right). (Bottom): Example trial sequence ans corresponding initial states following previous wins (triangles) or losses (circles) on right (R) or left (L) choices. Colors denote initial state biases, towards positive (blue) or negative (pink) bounds. **d** Effect of initial state values on psychometric curves. Colors same as **c**. Small deviations in initial state (grey) lead to largely horizontal biases whereas larger deviations (saturated colors) additionally reduce its effective slope (dotted black lines) or "sensitivity" to stimulus. **e** Pooling psychometric function (black) across trials with different initial state biases gives rise to apparent lapses (purple arrow). Conditioning the curve on previous rightward (blue) or leftward (pink) wins reveals a modulation of apparent lapses by trial history.

auditory decision-making task. Despite heterogeneity in history biases and lapse rates in this population, we show that a substantial fraction of lapses can be explained by the presence of history dependence during evidence accumulation. Further, our model predicts the time it takes to make decisions. We test these predictions in a novel task in rats with reaction time reports, and show that it captures patterns of choices, reaction times, and their history dependence. This model significantly improves our ability to predict the temporal dynamics of decision variables within and across trials in perceptual decision-making tasks, rendering choices that were previously thought to be stochastic, predictable.

## Results

### A common mechanism produces history biases and apparent lapses

It is often assumed that well-trained subjects in two-alternative forced choice (2AFC) tasks have faithfully learnt the likelihood function and priors that determine the structure of the task[23,41]. Under this assumption, the optimal decision-making strategy entails combining any knowledge about prior prevalence of available options with the stream of incoming evidence until a desired threshold of confidence is reached in favor of one of the options[41–43] (Fig. 1b top). This strategy converges to a drift-diffusion model (DDM) when evidence is sampled continuously[23]. In a DDM, one's belief about the correct option maps onto a diffusing particle that drifts between two boundaries, where the first boundary the particle crosses determines the decision (Fig. 1b). Correspondingly, the initial state of this particle encodes the prior belief, and the drift rate is set by the likelihood of incoming evidence (Fig. 1b). We refer to the evolving state of the particle in this model as 'accumulated evidence'.

However, in general, subjects may not know that the task structure is stationary, and might incorrectly assume that it is constantly changing[19]. In this case, even experienced subjects would not converge to a static estimate of prior probabilities and likelihood functions, but would instead continually update them from trial to trial. Here we consider choice behavior that results from non-stationary beliefs about priors, which result in trial-to-trial updates to the initial accumulator states. Although initial state updating is common to non-stationary beliefs in priors, likelihoods and reward functions, updates to the latter two additionally require drift rate updates (for a treatment of non-stationary likelihood functions which yield variability in drift rate, see[14,44]).

We assume that the initial state of the accumulator ($I$) is set based on the exponentially filtered history of choices and outcomes on past trials. Each unique choice-outcome pair (denoted by $h$; Fig. 1c) is tracked by its own exponential filter ($i^h$). On each trial $n$, each filter $i^h$ decays by a factor of $\beta^h$ and is incremented by a factor of $\eta^h$ depending

on the choice-outcome pair on the previous trial:

$$i^h(n) = \beta^h i^h(n-1) + \eta^h 1^h(o_{n-1}) \quad \text{where} \quad h = \{Rw, Lw, Rl, Ll\} \quad (1)$$

$\{Rw, Lw, Rl, Ll\}$ represent the possible choice-outcome pairs: right-win, left-win, right-loss, and left-loss respectively. $o_{n-1}$ is the choice-outcome pair observed on trial $(n-1)$ and $1^h(o_{n-1})$ is an indicator function that is 1 when $o_{n-1} = h$ and is 0 otherwise. The initial state of accumulation, $I$ on trial $n$ is given by the sum of these individual exponential filters:

$$I(n) = i^{Rw}(n) + i^{Lw}(n) + i^{Rl}(n) + i^{Ll}(n) \quad (2)$$

Such a filter can approximate optimal updating strategies under a variety of non-stationary beliefs. As an example, we show that this exponential filter can successfully approximate initial state updates during Bayesian learning of priors under the belief that the prior probabilities of the two hypotheses can undergo unsignaled jumps[5,19] (Supplementary Fig. 1). Nevertheless, we use this more flexible parameterization to allow for asymmetric learning from different choices and outcomes, which could be beneficial under generative models where one believes that one category persists for longer than another (requiring different decay rates), or correct and incorrect outcomes are not equally informative (requiring different update magnitudes). For instance, in a prior-tracking experiment where previous correct choices had a cumulative effect, but errors had a resetting effect[13], this could be captured in the exponential filter by faster decay rates for errors.

What are the consequences of such trial-by-trial updating of initial accumulator states for choice behavior? In a DDM, for a given initial state $I$ and drift rate $\mu$, the probability of choosing the option corresponding to bound $B+$ is given by:

$$P(B+) = \frac{1 - e^{-2\mu(B+I)/\sigma^2}}{1 - e^{-4\mu B/\sigma^2}} \quad (3)$$

where $B$ is the magnitude of the bound and $\sigma^2$ is the squared diffusion coefficient (derived from Palmer et al.[45]). The resultant psychometric curves for different values of initial accumulator states are plotted in Fig. 1d. This expression reduces to a logistic function of $\mu B/\sigma^2$ only when $I = 0$. Small deviations in the initial state largely resemble additive biases to the total evidence, shifting psychometric curves horizontally towards the option favored by the initial state. This corresponds to a change in the psychometric threshold i.e. the x-axis value at its inflection point (Fig. 1d lighter colors). Note that our use of the word "threshold" follows from Wichmann & Hill[27], referring to the x-axis value at the inflection point, whereas we refer to the slope at this inflection point as "sensitivity". Interestingly, large deviations in the

initial state produce qualitatively different effects on choices (Fig. 1d darker colors). They not only bias the choices towards the option consistent with the initial state but additionally reduce the effective sensitivity to evidence. This can be seen as reduction in slope at the inflection point of the psychometric curve (Fig. 1d dashed lines) in addition to a change in threshold. Therefore, trial to trial deviations in the initial state produce history-biased choices which have differently diminished dependence on the evidence.

The average choice behavior obtained by pooling choices with different history-biased initial states is a mixture of psychometric curves with varying thresholds and sensitivity to perceptual evidence. Such a psychometric curve is heavy-tailed[46,47] and appears to have asymptotic errors or "lapse rates" (Fig. 1e, black curve). These asymptotic errors are not truly evidence-independent, random decisions or true lapses, rather they are "apparent lapses" arising from evidence accumulation with deterministic history-based updates to the initial accumulator state. Importantly, these apparent lapses contribute to lapse rates when heavy-tailed psychometric curves are approximated by a logistic function. However, this approximation is bound to be inadequate if measurements were made for even higher stimulus strengths, making the heaviness of the tails even more evident. In such a setting, the psychometric curves obtained by conditioning on past trials' choice and outcome, or history-conditioned psychometric curves, are both horizontally and vertically shifted, i.e. they show history-dependent modulations in both threshold and lapse rate parameters (Fig. 1e, Supplementary Fig. 2b). Furthermore, trial-history modulated lapse rates are uniquely produced by history-biased initial accumulator states (and therefore reflect apparent lapses), in contrast to lapse rates observed in the unconditioned psychometric curve which might have additional extraneous causes[27,32,34], and therefore reflect both apparent and true lapses.

In this model, because history modulations of psychometric thresholds and lapse rates arise from one unified process, they are not allowed to vary independently of the decision-making process, or of each other. Rather their relative magnitudes are intimately coupled with and constrained by accumulation variables. For instance, increased magnitudes or timescales of initial state updating produce large fluctuations in the initial accumulator state across trials. This in turn reduces the effective sensitivity of the accumulation process to evidence, giving rise to more apparent lapses and history biases (Supplementary Fig. 2a). Similarly, changes in within-trial parameters of accumulation can dramatically influence these history modulations (Supplementary Fig. 2c). Decisions made with smaller accumulator bounds are more sensitive to initial state modulations, and therefore give rise to more apparent lapses and higher modulations of lapse rates and thresholds. Higher levels of sensory noise have a similar effect, yielding more apparent lapses, consistent with recent reports of lapse rates being modulated by sensory uncertainty[32]. Finally, impulsive integration strategies that overweigh early evidence rather than accumulating uniformly[23] exaggerate the influence of initial states, producing more apparent lapses and history biases.

Some definitions:
- *Lapse rate*: Lapse rates capture the difference between perfect performance and observed performance at the asymptotes, measured through sigmoidal fits to the psychometric curves.
- *True lapse*: A true lapse is a stochastic, evidence-independent choice that arises from cognitive processes entirely separate from the decision process, such as inattention or motor error.
- *Apparent lapses*: Apparent lapses are deterministic evidence-dependent choices, that nonetheless contribute to lapse rates when performance is averaged across trials.

## Rats display varying degrees of history-dependent threshold and lapse rate modulation

We sought to test if the comodulations posited by our model are present in rat decision-making datasets, in order to ascertain whether a unified explanation could underlie the links between history biases and lapses.

We first examined whether and how rat decision-making strategies were affected by trial history. We analyzed choice data from 152 rats (37522 ± 22090 trials per rat, mean ± SD; Supplementary Fig. 3a) trained on a previously developed task that requires accumulation of pulsatile auditory evidence over time ('Poisson Clicks' task[30]). In this task, the subject is presented with two simultaneous streams of randomly-timed discrete pulses of evidence, one from a speaker to their left and the other to their right (Fig. 2a). The subject must maintain fixation throughout the stimulus, and subsequently orient towards the side which played the greater number of clicks to receive a water reward. The trial difficulty, stimulus duration, and correct answer were set independently on each trial. Because this task delivers sensory evidence through randomly but precisely timed pulses, it provides high statistical power to characterize decision variables that give rise to the choice behavior.

Rats performed this task accurately (0.79 ± 0.04, mean accuracy ± SD, Supplementary Fig. 3b). Performance was stable with little to no change in accuracy across trials (mean slope ± SD across rats of linear fit to hit rate over trials: $1.13 \times 10^{-7} \pm 8.90 \times 10^{-7}$; Supplementary Fig. 3c) reflecting asymptotic behavior rather than task acquisition. Rats showed history dependence in their choices, largely tending towards a "win-stay, lose-switch" dependence (Supplementary Fig. 3e). We found substantial individual variability in the dependence of rats' choices on history in the dataset. Some rats were weakly influenced by history (Fig. 2b left) while others showed a history-dependent modulation of the psychometric threshold parameter (Fig. 2b middle) or a history-dependent modulation of both threshold and lapse rate parameters (Fig. 2b right). The population as a whole most closely resembles Example rat 3, with both threshold and lapse rate parameters being significantly different following left and right wins while sensitivity is not affected ($p = 0.8$ for sensitivity, $3 \times 10^{-17}$ for bias, $8 \times 10^{-8}$ for left lapse, $6 \times 10^{-7}$ for right lapse, two-sided Mann-Whitney U-test, $n = 152$ Fig. 2c). Using simulations, we confirmed that the logistic fits to psychometric curves can reliably recover performance asymptotes i.e lapse rates particularly in the parameter regimes of this dataset (Supplementary Fig. 4). As predicted by our model (Fig. 1e), trial-history biased both threshold and lapse rate parameters in the same direction (e.g. both biased toward rightward choices following right rewards). Moreover, the vast majority of rats show comodulations of both parameters by history (Pearson's correlation coefficient: $r = -0.35$, $p = 7.28 \times 10^{-6}$; Fig. 2d). Across rats, on average 17 ± 12% of lapses are modulated by trial history and therefore could potentially reflect apparent rather than true lapses (Supplementary Fig. 3d). These findings support the conclusion that rat decision-making strategies, while idiosyncratic, largely show history-dependent effects consistent with our model. Next, we tested the model more directly using trial-by-trial model fitting.

## History-dependent initial states capture comodulations in thresholds and lapse rates in the data

To test whether the observed history modulations in thresholds and lapse rates arise from trial-by-trial updates to the initial accumulator state, we extended an accumulator model previously adapted to this pulsatile task[30] to incorporate History-dependent Initial States (abbreviated as HISt, Fig. 3a). As before, we model this history-dependence using an exponential filter over past trials' choices and outcomes (Fig. 1c). Hence, across trials the accumulator model with HISt produces apparent lapses, as well as coupled history modulations in psychometric threshold and lapse rate parameters.

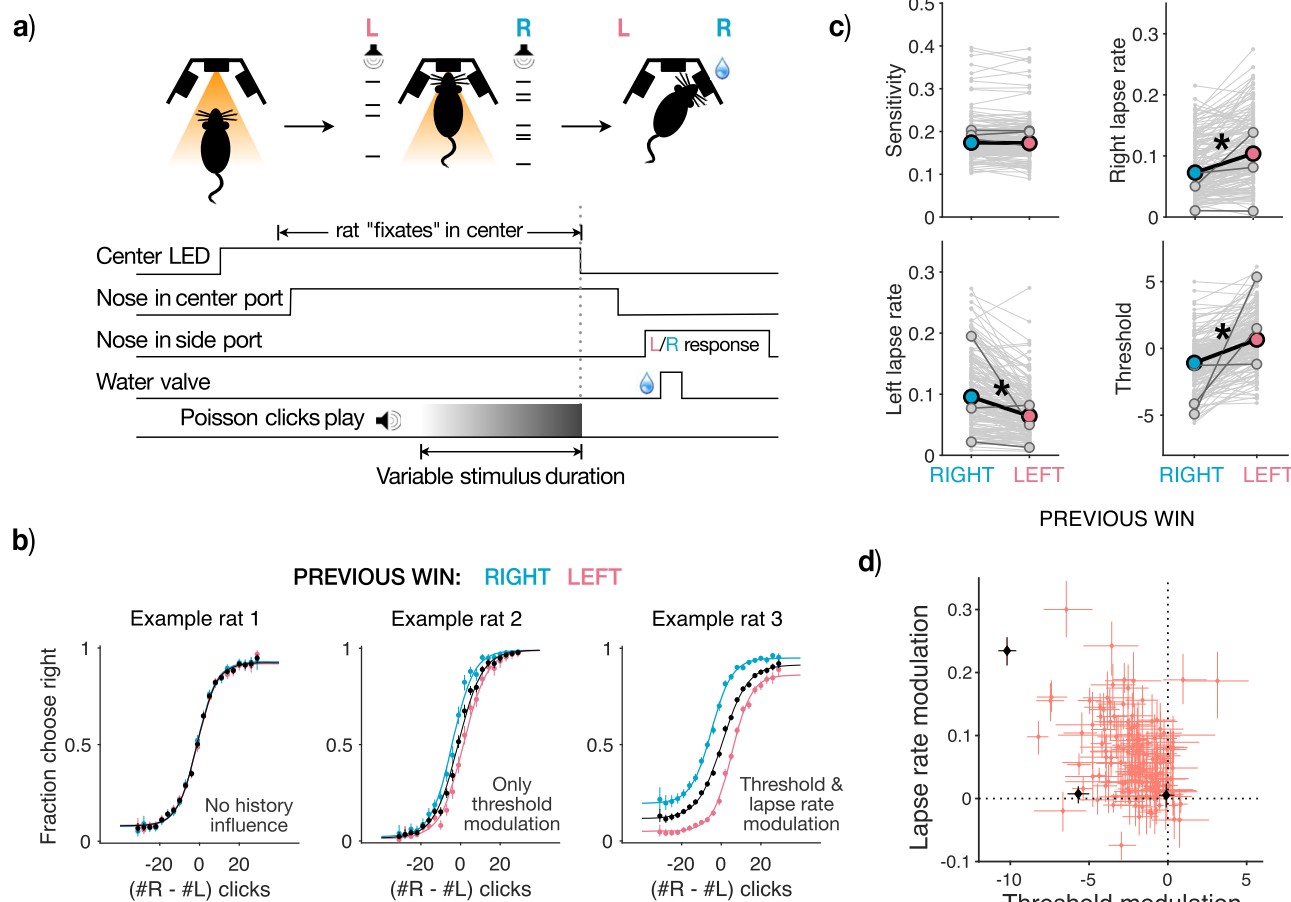

**Fig. 2 | History-dependent threshold and lapse rate modulations in a large-scale rat dataset. a** Schematic of evidence accumulation task in rats: (Top): Phases of the 'Poisson clicks' task, including trial initiation in center port (left), evidence accumulation based on two streams of Poisson-distributed auditory clicks (middle) and choice report in one of two side ports followed by water reward for correct choices (right). (Bottom): Time-course of trial events in a typical trial. (Figure adapted with permission from Bingni W. Brunton et al., Rats and Humans Can Optimally Accumulate Evidence for Decision Making. Science 340,95-98(2013). DOI:10.1126/science.1233912) **b** Individual differences in history-dependence: Psychometric functions of three example rats from a large-scale dataset, displaying different kinds of history modulation. Choices are plotted conditioned on previous left (blue), right (pink) or all wins (black). (Left): Example rat with no history-dependence in choices, resembling the ideal observer. (Middle): Example rat with modulations of the threshold parameter alone, resembling the dominant conceptualization of history bias. (Right): Example rat with history-dependent modulation of both threshold and lapse rate parameter, similar to the majority of the population. Errorbars represent 95% binomial confidence intervals

around the mean (n = [16946, 20577, 37523] trials for example 1, [8568, 9549, 18117] trials for example 2, [29358, 30821, 60179] trials for example 3 for psychometric curves conditioned on [right, left or all wins]) **c** Dataset displays significant modulations of both threshold and lapse rate parameters: Scatters showing parameters of psychometric functions following leftward wins (post left, blue) or rightward wins (post right, pink). Each pair of connected gray points represents an individual animal, solid colored dots represent average parameter values across animals. Trial history does not significantly affect the sensitivity parameter (top left) but significantly affects left, right lapse rate and threshold parameters (top right and bottom panels). ($p = 0.8$ for sensitivity, $3 \times 10^{-17}$ for bias, $8 \times 10^{-8}$ for left lapse, $6 \times 10^{-7}$ for right lapse, two-sided Mann-Whitney U-test, $n = 152$) **d** Scatter comparing threshold and lapse rate modulations in the entire population ($n = 152$). Each dot is an individual animal, best-fit parameter values ± 95% bootstrap CIs. Black points represent example rats. The majority of the population lies in the top left quadrant, showing comodulations of both threshold and lapse rate parameters by history.

Within a trial, our accumulator model leverages knowledge of the timing of each evidence pulse to model the sensory adaptation process as well as to estimate the noise and drift of the accumulator variable (Fig. 3a top bubble, Methods). The model includes a feedback parameter that controls whether integration is leaky, perfect, or impulsive. Following Brunton et al.[30], this model also includes (biased) random choices independent of the accumulator value on a small fraction of trials ($\kappa$) - we consider decisions arising from this process to be "true lapses" because they are evidence-independent, unlike apparent lapses which still retain some evidence-dependence (Fig. 3a bottom bubble).

We performed trial-by-trial fitting of the accumulator model with and without History-dependent Initial States (HISt) to choices from each rat using maximum likelihood estimation (Methods). We find that the accumulator model with HISt captures both psychometric curve threshold and lapse rate modulations well across different regimes of

rat behavior, as evident from fits to example rats (Fig. 3b). Moreover, conditioning rats' psychometric curves on model-inferred initial state values reveals that the initial state captures a large amount of variance in choice probabilities (Fig. 3c), resembling theoretical predictions (Fig. 1c). This shows that the initial state is a key explanatory variable underlying choice variability both across and within individuals, that jointly modulates multiple features of the empirical psychometric curves in a parametric fashion. We used Bayes Information Criterion (BIC) to determine whether adding HISt to the accumulator model was warranted (Fig. 3d, e). Individual BIC scores recommended that adding HISt was warranted in 147/152 rats (Fig. 3d). This model also best captured choices across the population as a whole, with significantly lower mean BIC scores across rats (Mean per trial BIC score for HISt: $0.91 \pm 0.01$ vs. no HISt: $0.93 \pm 0.01$, $p = 9.85 \times 10^{-18}$, paired t-test; Fig. 3e). Next, we compared the psychometric threshold and lapse rate

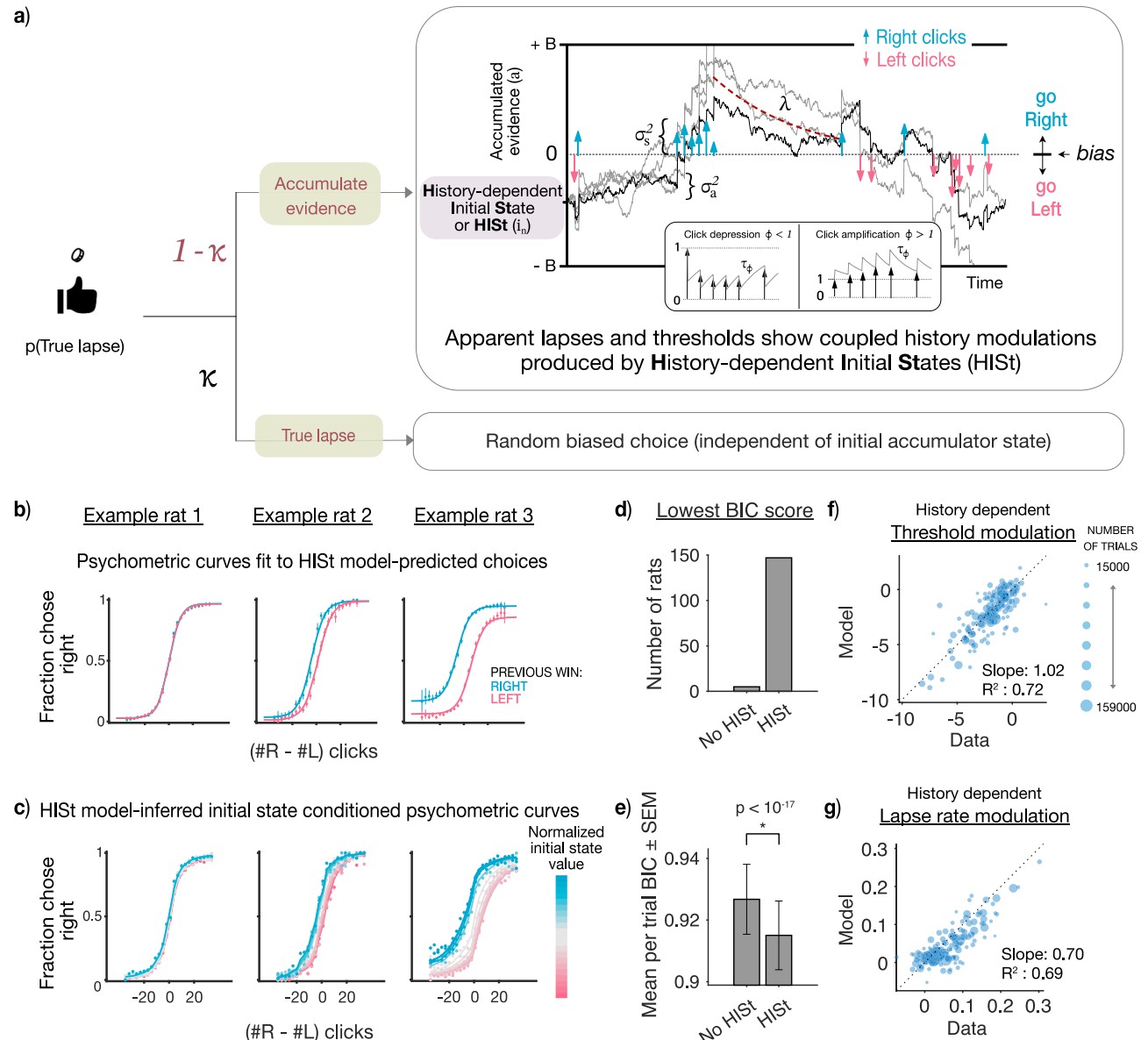

**Fig. 3 | History-dependent initial states capture comodulations in thresholds and lapse rates in the data. a** Schematic of the model used to fit rat data in the Poisson Clicks task. **(Top)**: The model consists of trial history-dependent initial states (HISt) that can produce history-dependent apparent lapses and threshold modulations. Additionally, the model consists of sensory noise ($\sigma_s^2$) in click magnitudes, adaptation of successive click magnitudes based on an adaptation scale ($\phi$) and timescale ($\tau_\phi$), accumulator noise ($\sigma_a^2$), leak in the accumulator ($\lambda$), and decision bounds +/−B[30]. **(Bottom)**: On $\kappa$ fraction of trials, the model chooses a random action with some bias ($\rho$) reflecting motor errors or random exploration. These true lapses are not modulated by history, such that any history modulations arise from the initial states alone. (Figure adapted with permission from Bingni W. Brunton et al., Rats and Humans Can Optimally Accumulate Evidence for Decision Making. Science 340,95-98(2013). DOI:10.1126/science.1233912) **b** Model fits to individual rats: Psychometric data (mean accuracy ± 95% binomial confidence intervals) from 3 example rats conditioned on previous rightward (blue) or leftward (wins),

overlaid on model-predicted psychometric curves (solid line) from the accumulation with HISt model. ($n$ = [16946, 20577] trials for example 1, [8568, 9549] trials for example 2, [29358, 30821] trials for example 3 for psychometric curves conditioned on [right, left wins]) **c**: Psychometric curves (solid line) from the same example rats conditioned on model-inferred initial states (colors from pink to blue).
**d** Distribution of best fitting models for individual rats: **e** Model comparison using BIC by pooling per trial BIC score across rats and computing mean ($n$ = 152). Mean of per trial BIC scores across rats were significantly lower for model with HISt ($p$ = $9.85 \times 10^{-18}$, one-sided paired t-test) indicating better fits. Error bars are SEM. For individual data points see Supplementary Fig. 6f Individual variations in history modulations captured by the accumulator model with HISt: History modulations of threshold parameters measured from psychometric fits to the raw data (x-axis) v.s. model predictions (y-axis). Individual points represent individual rats ($n$ = 152), point sizes indicate number of trials. **g** same as (**f**) but for history-dependent lapse rate modulations.

modulations produced by this model to the modulations in the data, as determined by conditioning the psychometric functions on trial-history (Fig. 3b). As predicted, the model successfully accounted for modulations in both these distinct psychometric features via the singular process of trial-by-trial history-dependent updates to the initial accumulator state. Next, we examined the extent to which these modulations were captured across individual rats (Fig 3f, g). We

quantified these history modulations as follows: "threshold modulations" are defined as the horizontal distance between the midpoints of psychometric curves conditioned on previous wins and losses, and "lapse rate modulation" as the vertical distance between the asymptotes of these curves (Methods: History modulation of psychometric parameters, also see Supplementary Fig. 2b). This was done separately for model-predicted and rat choices and then compared. Across

individuals, the model with HISt captured a substantial amount of variance [$R^2 = 0.72$ (threshold parameter), $R^2 = 0.69$ (lapse rate parameter)] and showed good correspondence to the empirical modulations in data [slope = 1.02 (threshold parameter), slope = 0.70 (lapse rate parameter)].

In our model, apparent lapses show history modulations since they are produced by history-dependent initial accumulator states, while true lapses do not since they result from an occasional flip in the final choice and are independent of the accumulator value (following Brunton et al.[30]). Such kinds of true lapses could reflect errors in motor execution or random exploratory choices made despite successful accumulation (Supplementary Fig. 5b). However true lapses could also occur due to inattention, i.e. an occasional failure to attend to the stimulus. In such cases, the optimal strategy devoid of sensory evidence is to deterministically choose the side favored by the initial accumulator state (Supplementary Fig. 5c). Therefore, inattentional true lapses, while remaining evidence independent, may nevertheless be modulated by history due to their initial state dependence. In order to account for this possibility, we fit an additional "inattentional" variant of the accumulator model with HISt (Supplementary Fig. 5a, c), and found that it was closely matched on BIC scores with the previous model which we label as the "motor error" variant (Supplementary Fig. 5e, f). Moreover, the inattentional variant, which additionally allows true lapses to depend on history, only captured slightly more variance in history modulations of lapse rates, at the expense of history modulations of thresholds (Supplementary Fig. 5d) while a variant of the model with inattentional true lapses but without HISt failed completely to capture the comodulation and performed much worse overall (Supplementary Fig. 6). Together these two findings support the hypothesis that apparent lapses produced by history-dependent initial states (rather than true lapses due to motor error or inattention) are the major driver of history-dependent comodulations in psychometric thresholds and lapse rates in the dataset.

To gain further insight into the initial state updating dynamics, we examined the fit parameters controlling the magnitude and timescale of updates (Supplementary Fig. 7). We found that across the population of rats, updates following wins and losses had similar magnitudes, but opposite signs, suggesting a tendency to repeat after wins and switch after losses. We compared these fits to those from a restricted version of the model whose initial state dynamics correspond to optimal updates in a Dynamic Belief Model[48] (Supplementary Fig. 1) and found that about a third of the population (47/152 rats) were consistent with this form of statistical inference (Supplementary Fig. 7b). The remainder of the population did not show a significant correlation between post-win and post-loss parameters, consistent with a statistical model that treats wins and losses differentially[13,49] (Supplementary Fig. 7c).

To summarize, our model predicted that the initial accumulator state should be the underlying variable that jointly drives history-dependence in thresholds and lapse rates – implying that our accumulator model with HISt should be able to simultaneously capture variability in both these parameters across rats. Our rat dataset strongly supports this prediction, lending evidence to the hypothesis that history-dependent initial states give rise to apparent lapses, and are the common cognitive process that underlie links between these two sub-optimalities that were previously thought to be distinct from each other.

**Reaction times support history-dependent initial state updating**
In our model with history-dependent initial accumulator states, the time it takes for the accumulation variable to hit the bound determines the duration that the subject deliberates for, before committing to a choice. Therefore in addition to choices the model makes clear predictions about subjects' reaction times (RTs). We sought to test if these predictions are borne out in subject RTs.

To this end, we trained rats ($n = 6$) on a new variant of the auditory evidence accumulation task, with two key modifications that allowed

us to collect reaction time reports (Fig. 4a). First, in this new task the stimulus is played as long as the rat maintains their nose in the center port (or "fixates") and stops immediately when this fixation is broken. Second, in this task the rat has to correctly report which speaker's auditory click train is sampled from a higher Poisson rate to receive a water reward (unlike the non-reaction time task where the subject has to report the side which played the greater number of clicks). Rats perform this task with high accuracy (Fig. 4b left panel, average accuracy: $0.75 \pm 0.02$, number of trials $37205 \pm 14247$, mean ± SD). Similar to the previously analyzed data, their choices are impacted by recent trial history (Fig. 4b right panel). Moreover, trial-history dependent modulation of psychometric function parameters (Fig. 4c) resembles that of the non-reaction time task (Fig. 2c; $p = 0.69$ for sensitivity, 0.004 for threshold, 0.02 for left lapse rate, 0.02 for right lapse rate, Mann-Whitney U-test). Once again, this history modulation of both psychometric threshold and lapse rate parameters in tandem is consistent with our singular accumulator model with history-dependent initial states.

Moreover, RTs of these rats display several signatures predicted by our model (Fig. 4d–f). First, trial-to-trial variability in the initial state of the accumulator is expected to give rise to shorter RTs on error trials compared to correct trials[22] (Fig. 4e, left). This is because trials in which the initial state is closer to the incorrect bound are more likely to be errors, but because of the closer bound they are also likely to hit it faster. This is unlike a standard DDM with no trial-to-trial variability in parameters, where RTs for correct and error trials are of similar magnitudes (Fig. 4d, left). Indeed in the rat dataset, error RTs are consistently shorter than correct RTs across rats (Fig. 4f, left). Second, initial state updates towards previously rewarded choices (such as in a win-stay agent) are expected to produce shorter RTs when the current stimulus favors the previously rewarded choice[19,24] (Fig. 4e, middle). We find that this signature is also present in the dataset across rats (Fig. 4f, middle). Finally, variability in the initial state is most influential early in the decision process, predicting that the majority of history dependence in choices occurs on trials with fast RTs[12] (Fig. 4e, right). Indeed, the data displays this pattern as well, with repetition bias being most prominent for short RTs, disappearing and turning into a weak alternation bias for long RTs (Fig. 4f, right). Taken together, these three signatures offer strong, complementary evidence from RTs for the prevalence of history-dependent initial states in rats performing this evidence accumulation task.

We directly test if our model can simultaneously capture reaction time patterns and history-modulation of psychometric threshold and lapse parameters by jointly fitting choices and RTs of individual subjects in a trial-by-trial fashion (see Methods). We find that the history-dependent initial state model jointly captures patterns of choices, reaction times, and their history modulations in the data (Fig. 4g - fits from example rat, Supplementary Fig. 8 - fits from all rats). This model accounts for substantial variance in history-dependent threshold and lapse rate modulations (Fig. 4h). We also fit a hybrid variant of the accumulator model with HISt that flexibly allows true lapses to be motor-error like and unaffected by history, or inattention-like and additionally be modulated by history (Supplementary Fig. 9a, b). While this model has a better BIC and leads to a slight improvement in correspondence to the history modulation of psychometric lapse rates, it does so at the cost of correspondence to modulations in psychometric thresholds (Supplementary Fig. 9c–e). This equivocal improvement over the HISt model in capturing the threshold and lapse rate modulations support the conclusion that HISt and its resultant apparent lapses (rather than true lapses) are a major contributor to the observed comodulation of both parameters.

Overall, these results show that the history-dependent initial state updates that we invoked to explain apparent lapses in rodent data are corroborated by their reaction times, and accounting for them can help render a sizable fraction of decisions — that would have been

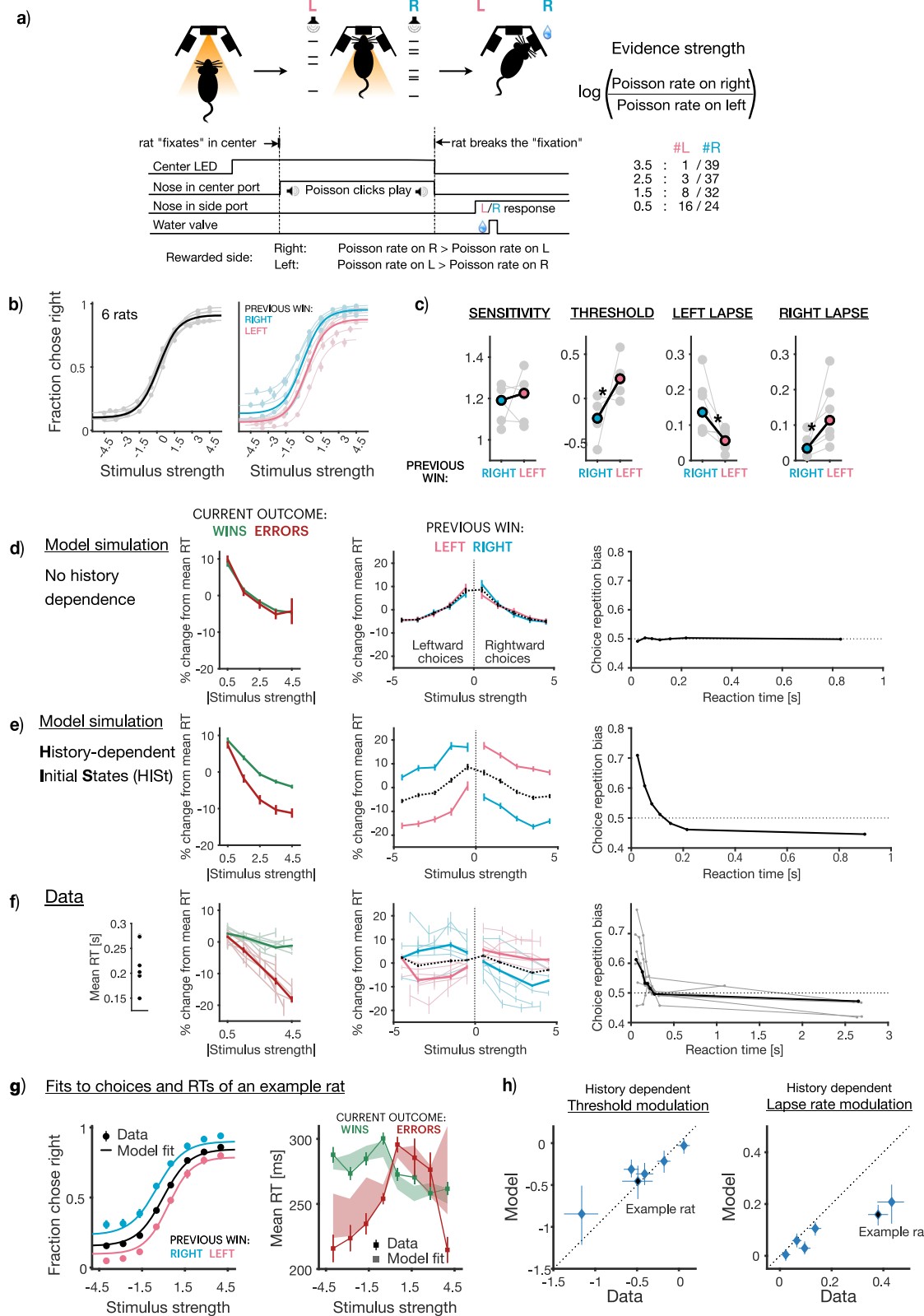

## Discussion

History biases and lapses have both long been known to impact perceptual decision-making across species. However, they have largely been assumed to be distinct from each other, despite their frequent co-occurrence and comodulation. Here, we propose that normative accumulation under misbeliefs of non-stationarity can produce both history biases and apparent lapses, offering an explanatory link between the two suboptimalities. This corresponds to history-dependent trial-to-trial updates to the initial state of an evidence accumulator. We show that such updates produce choices with varying biases in psychometric thresholds as well as varying sensitivities to

**Fig. 4 | Model predictions about reaction times are borne out in data.**
**a** Schematic of reaction time task in rats (Figure adapted with permission from Bingni W. Brunton et al., Rats and Humans Can Optimally Accumulate Evidence for Decision Making. Science 340,95-98(2013). DOI:10.1126/science.1233912) **b** Average choice behavior on all trials (left; n = 223231 trials) and following previous right (n = 86109 trials) or left wins (n = 82678 trials; right) across 6 rats (solid line), overlaid on individual rat behavior (translucent lines). Errorbars represent 95% binomial confidence intervals around the mean. **c** Average parameters (solid points) of history-conditioned psychometric curves, overlaid on individual parameters (translucent points) showing significant history modulations in threshold and lapse rate parameters ($p = 0.69$ for sensitivity, 0.004 for threshold, 0.02 for left lapse rate, 0.02 for right lapse rate, two-sided Mann-Whitney U-test; $n = 6$). **d–f** Reaction time signatures (**d**) expected from accumulator models with no history dependence in initial states, (**e**) expected from accumulator models with history-dependent initial states and (**f**) observed in data ($n = 223{,}231$ trials across all stimulus strengths and rats). (Leftmost column) error reaction times are expected to be shorter if initial states are history-dependent. Red (green) represents RTs on errors (wins). (Middle column) reaction times on trials following right wins (blue) are expected to be lower on rightward stimuli (positive half of x-axis), and similarly following left wins (pink). (Rightmost columns) repetition biases in choices are expected to occur more frequently for short reaction times, when the effect of initial states is strong. Error bars represent SEM. **g** Joint fits of the accumulator model with history-dependent initial states to choices (left) and reaction times (right) of an example rat ($n = 24413$ trials). Data represented by points (circles: choices, mean accuracy ± 95% binomial confidence intervals; squares: reaction times, mean RT ± SEM) and model fits represented by lines (choices) or shaded bars (reaction times, thickness represents 95% bootstrap prediction intervals). Reaction times (right) are split by wins (green) or errors (red). **h** Scatter plot showing correspondence between history modulations in threshold (left) or lapse rate (right) parameters derived from data (x-axis) and model fits (y-axis). Individual points represent individual rats ($n = 6$), best-fit parameter values ± 95% bootstrap CIs.

evidence, yielding apparent, history-modulated lapse rates when choices are averaged across trials (Fig. 1). Our model postulates that the initial state of the accumulator is a key underlying variable that jointly modulates psychometric thresholds and lapse rate parameters, with the exact nature of this comodulation determined by the within and across trial parameters governing evidence accumulation. We tested this model in a large rat dataset consisting of choices from 152 rats (Fig. 2) and confirmed its predictions using detailed model-fitting. We found that the singular process of history-dependent initial states successfully captured a substantial amount of variance in history modulations of both thresholds and lapse rates in the dataset (Fig. 3). Finally, we tested the reaction time predictions of the model in a novel task in rats, and confirmed that the data showed signatures of initial state updating. The model could successfully capture choices, reaction times, and history modulations in psychometric thresholds and lapse rates (Fig. 4). Altogether, our results suggest that history biases and a substantial amount of variance attributed to lapses may reflect a common mechanistic process, whose evolution can be precisely tracked both within and across trials.

History biases in perceptual decision making tasks have been modeled using initial state updates to DDMs in humans and non-human primates[2,5,24]. These studies tended to have relatively small magnitudes of history bias, and miniscule lapse rates, hence being well captured by small deviations in the initial state of a DDM, which largely yield horizontal shifts in the psychometric function. This regime of initial state updates is well approximated by a logistic function with additive biases, which is the dominant descriptive model used to characterize history-dependent psychometric curves[3,4,6,8,9,11–13,17,26,34,50]. However, as we demonstrate, when deviations in the initial state are large, this logistic approximation breaks down. This fact has been overlooked in much of the literature. Consequently, even in datasets with large history biases and lapses, the logistic formulation continues to be favored[9,17,18,34], albeit requiring additional components. Such effects tend to be prevalent in rodents but not human or non-human primate behavior. Our demonstration predicts that the full range of initial state effects should resemble concurrent, trial-by-trial changes in both threshold and sensitivity parameters of the logistic function. Indeed, Ashwood et al.[34] found that apparent lapses in several rodent datasets can be better captured by runs of trials with such concurrent modulations, yielding biased "disengaged" states. Our model captures both these behavioral regimes simply using different magnitudes of initial state updates, rendering it capable of accounting for individual differences across animals, and potentially even species with very different behavioral signatures, as long as the constraints between initial state updating, history biases and lapses are obeyed.

A number of previous studies have hinted at the performance-limiting effect of sequential biases, variability in initial points and/or sensitivity across trials[23,46,47]. Nguyen et al.[47] examined the optimal decision making strategy under a non-stationary generative model, and arrived at psychometric curves similar to the heavy-tailed curves produced by our model. Similarly, Shen et al.[46] examined decision-making under variable "precision" across trials, which also yields heavy-tailed psychometrics, trading off against lapse parameters. However, to our knowledge, ours is the first study to directly examine the effect of sequential biases on lapse rates, and link the two relatively separate literatures. Our model formulation shares some features with previous work on sequential biases, albeit with some distinct features - our model is a Drift Diffusion Model with history-dependent initial states (similar to Nguyen et al.[47], but unlike Kim et al.[25], who use an adaptive LATER model) adapted to discrete stimuli for the purpose of trial-by-trial modeling. Our model's initial states are a continuous variable, unlike Urai et al.[12], whose initial states take on one of two possible discrete values. Also, our model's initial states are set by a flexible exponential filter on several past choices and outcomes, unlike Nguyen et al.[47], Kim et al.[25], Yu et al.[48] and other variants of the Dynamic Belief Model, albeit reducing to them for certain restricted parameter regimes.

In our treatment, we only considered history-dependent updates to the initial state of a DDM. Such a mechanism is normative under non-stationary beliefs about the prior (note that this is the case if the agent assumes that a shift in the prior over stimulus categories maps onto an overall shift in the prior over stimulus difficulties – see Drugowitsch et al.[44] for a detailed treatment), which is our favored interpretation as it aligns with other studies of history biases[2,8,19,20,24,25,51,52]. Nevertheless, these updates may also reflect other heuristic strategies[53] which we accommodate using our flexible parameterization of initial state updates. Animals may entertain non-stationary beliefs about other elements of the decision process, such as the rewards or likelihoods[14,15,32,42]. Normative updating in such situations still reduces to initial state updates in simple settings (for e.g. non-stationary rewards for a single difficulty[54,55]), but in more complex ones it affects drift rates or bounds in addition to initial states[12,14,44,45,56–58]. This commonality of initial state updating to many different non-stationary beliefs motivated us to probe its role in producing apparent lapses, and indeed this mechanism was able to explain an impressive amount of variance in our dataset, leading us to conclude that initial state updating is at least a major factor driving animal behavior. Another crucial possibility is trial-to-trial variability in drift rates, which is known to give rise to longer error RTs than correct RTs[43,59–61] and is a signature often reported in monkeys and humans[62,63]. We did not observe the reaction time signatures of drift rate variability in our dataset, instead we identified signatures of initial state variability, where error RTs were shorter than correct RTs, rather than longer. However, drift rate updates may represent an alternative mechanism through which history-modulated apparent lapses could occur in other datasets. It is worth noting that certain task designs include

efforts to actively measure and counter trial history biases. In such cases, lapses may still occur, likely due to exploration or inattention. In this manuscript, we refer to lapses caused by these factors as "true lapses", since they cannot be explained by fluctuations in DDM-related parameters.

Lapse rates are often considered to be a mixed bag comprising several different noise processes separate from the decision process, yet most studies so far have focused on one or more of these component processes in isolation[32,34]. In this work, we have attempted a more expansive approach of considering multiple processes at once, in an attempt to partition lapse rate variance into mixtures of deterministic and stochastic components. We distinguished apparent lapses that interact with sensory evidence from two models of "true" lapses that are both evidence independent − motor error or exploration, which does not interact with the accumulator, and inattention, which may still depend on its initial state. While we find that the behavior of our rats is best described by a mixture of apparent lapses and the two true lapse variants, it is primarily the apparent lapses (rather than either true lapse variant) that capture the links between the suboptimalities i.e. the history-dependent comodulations in psychometric thresholds and lapse rates. A previous study proposed an evidence-dependent model of true lapses, uncertainty-guided exploration[32], in order to account for the scaling of lapse rates with sensory noise. Although we don't explicitly consider this model, our model of apparent lapses already displays this property, with higher levels of sensory noise leading to more frequent apparent lapses.

Our model predicts that an increased reliance on history (i.e., larger shifts of the initial states) should produce more apparent lapses. Indeed, this could provide an explanation that links disparate sets of observations from previous studies: while some studies have reported that perturbations of secondary motor cortex and striatum give rise to higher lapse rates[32,36–39], others have shown that the effects of perturbing these regions seems to resemble an increased history-dependence[39,64]. Interpreting these results through the lens of our model, we would conclude that these regions play a crucial role in the interaction of history-dependent initial states with sensory evidence, making them a potential common neural substrate that could contribute to both kinds of suboptimalities. Indeed, increased history dependence upon M2 perturbation has been shown to be mediated by increased bias in the initial value of the neurally derived accumulator variable[64]. Similarly, DMS perturbations had large effects on lapse rates in moderately engaged behavioral states that were influenced by both sensory evidence and history[50]. Our model could also help explain why Busse et al.[3] found that mice with higher lapse probabilities showed higher history dependence, or results from IBL[18] who observed a modulation in lapse rates in addition to horizontal biases upon explicit manipulation of category priors. Nonetheless, these observations do not preclude the possibility that there are indeed independent neural mechanisms and/or areas through which trial-history effects and lapses (particularly true lapses) arise. Consistent with this, studies have implicated different brain areas in producing deterministic vs stochastic biases in action timing[65], and even different sub-circuits within the same area in giving rise to distinct behavioral strategies[66]. Detailed manipulations of brain regions with prior information such as in studies like IBL (2023)[67] could help pinpoint the neural mechanisms through which these suboptimalities arise.

One interesting future line of investigation is to probe the precise nature of the model of non-stationarity over priors assumed by animals in such tasks. The range of parameter values inferred using our flexible formulation could offer a useful starting point for this line of investigation. For instance, Dynamic Belief Models[19,68], a popular class of generative models over priors, correspond to a narrowly constrained set of parameter values in our model. Such an understanding would not only afford more reliable control of behavior and more accurate interpretation of neural correlates in stationary tasks, but

could also yield insight into the inductive biases that allow animals to learn quickly and efficiently in non-stationary, naturalistic settings.

## Methods
### Subjects
Animal use procedures were approved by the Princeton University Institutional Animal Care and Use Committee (IACUC #1853). All subjects ($n = 152$) were adult male Long Evans rats, typically housed in pairs. Housing both male and female rats in our rodent system resulted in a significant rise in aggression especially in certain transgenic rat lines to the point of making these rats unsafe to handle. This prevented us from studying both sexes and including sex as a factor in our study design. Rats that trained during the day were housed in a reverse light cycle room. Rats were typically aged between 6-24 months. Rats had free access to food but in order to to motivate them to work for water reward, they were placed on a controlled water schedule: 2-4 hours per day during task training, usually 7 days a week and between 0 and 1-hour ad lib following training.

### Drift diffusion model of decision-making
We use a standard formulation of sequential decision-making[23,43], in which an agent is faced with a stream of noisy sensory evidence $\epsilon_{1:t}$ coming from one of two hypotheses $H_1$ and $H_2$. The agent has to decide between sampling for longer or choosing one of two actions $L,R$ (reaction time regime) or has to choose one of two actions after a fixed amount of evidence (fixed duration regime). Such a problem can be formulated as one of finding an optimal policy $\pi_t$ in a partially-observable markov decision process[43,69], whose solution can be written as a pair of thresholds on the log-posterior ratio $\log\left(\frac{g(t)}{1-g(t)}\right)$, where $g(t) = p(H_1|\epsilon_{1:t})$:

$$\pi_t = \begin{cases} \text{choose L,} & -B \geq & \log\left(\frac{g(t)}{1-g(t)}\right) \\ \text{sample,} & -B < & \log\left(\frac{g(t)}{1-g(t)}\right) < B \\ \text{choose R,} & & \log\left(\frac{g(t)}{1-g(t)}\right) \geq B \end{cases} \quad (4)$$

The log posterior ratio can be further broken down into a sum of log prior ratios and log-likelihood ratios, using Bayes rule:

$$\log\frac{p(H_1|\epsilon_{1:t})}{p(H_2|\epsilon_{1:t})} = \log\frac{p(H_1)}{p(H_2)} + \log\frac{p(\epsilon_{1:t}|H_1)}{p(\epsilon_{1:t}|H_2)} \quad (5)$$

The optimal policy can equivalently be expressed in terms of the prior and sum of momentary sensory evidence $x(t) = \sum_t \epsilon_t$, which are sufficient statistics of the posterior[43,70]. In the continuous time limit, when the average rate of evidence increments or drift rate is $\mu$, and the standard deviation of sensory noise is $\sigma$, this corresponds to a drift-diffusion model that terminates when it reaches one of two bounds[23] and whose initial state $I$ is proportional to the log prior ratio:

$$dx = \mu dt + \sigma dW, \quad x(0) = I = k \cdot \log\frac{p(H_1)}{p(H_2)} \quad (6)$$

In this case, the probability of choosing rightward actions, i.e. hitting the upper bound can be written analytically as follows (derived from ref. 45):

$$P(B+) = \frac{1 - e^{-2\mu(B+I)/\sigma^2}}{1 - e^{-4\mu B/\sigma^2}} \quad (7)$$

In cases where trial difficulties (and hence drift rates) vary from trial to trial the optimal policy includes time-dependent, collapsing bounds on the posterior. However, under certain circumstances, constant bounds on $X_t = \sum_t \epsilon_t$ implement close-to-optimal collapsing

bounds on the posterior[43,71], which is the regime we assume for our analysis.

## Models of initial state updating

We model initial state updating as a sum of exponential filters over past choice-outcome pairs (*Rw*: right-wins, *Lw*: left-wins, *Rl*: right-loss, *Ll*: left-loss). So the initial state *I* at trial $n + 1$ is given by:

$$I(n+1) = i^{Rw}(n+1) + i^{Lw}(n+1) + i^{Rl}(n+1) + i^{Ll}(n+1) \tag{8}$$

where each filter $i^h$ decays by a factor of $\beta^h$, and is incremented by a factor of $\eta^h$ following the observation of that particular choice-outcome pair, i.e

$$i^h(n+1) = \eta^h 1^h(o_n) + \beta^h i^h(n) \quad \text{where} \quad h = \{Rw, Lw, Rl, Ll\} \tag{9}$$

$o_n$ is the choice-outcome pair observed on trial *n* and $1^h(o_n)$ is an indicator function that is 1 when $o_n = h$ and is 0 otherwise.

For non-reaction time datasets, in order to ensure good identifiability we constrained the update parameters to be the same following both left and right losses i.e. $\beta^h$ and $\eta^h$ to be the same for $h = \{Rl, Ll\}$. Additionally, following correct trials, we enforce the timescale of update i.e. $\beta^h$ to be the same for left and right trials $h = \{Lw, Rw\}$ while allowing the increment parameters $\eta^h$ to be different. When, $\beta^h$ and $\eta^h$ are the same $\forall\, h$, this rule reduces to an approximation of the Bayesian update for the Dynamic Belief Model[19], which tracks a prior that undergoes discrete unsignaled switches at a fixed rate. We compared this reduced (DBM) model to the exponential filter as described above (Supplementary Fig. 6a, b). While model comparison revealed that not every rat required all parameters to be different, the unconstrained model is the most general form that best captures behavior across rats.

## Psychometric curves

Psychometric curves model the probability of a subject choosing one of the options (e.g. right) as a function of stimulus strength. We parametrize the psychometric curve as a 4-parameter logistic function:

$$P(\text{choose Right}) = \kappa_0 + \frac{\kappa_1}{1 + e^{-b(x - x_0)}} \tag{10}$$

where $x_0$ is the threshold parameter that additively biases the stimulus *x*, *b* measures sensitivity to the stimulus, $\kappa_0$ is the left asymptote or left lapse rate and $\kappa_1$ scales the logistic function. Therefore, the right asymptote is given by $\kappa_0 + \kappa_1$ and the right lapse rate itself is given by $1 - (\kappa_0 + \kappa_1)$. We fit all four of these parameters $\{\kappa_0, \kappa_1, x_0, b\}$ to choices generated by either the DDM (Fig. 1), rats (Figs. 2–4), or accumulator models adapted to the tasks (Figs. 3, 4) using a gradient-descent algorithm (interior-point) to maximize the (Binomial) log likelihood of choices using MATLAB's constrained optimization function *fmincon*. $\kappa_0$ and $\kappa_1$ were both constrained to lie within the interval [0, 1]. 95% confidence intervals on these parameters were generated using bootstrapping. Throughout this manuscript, we follow the convention from Wichmann and Hill (2001) and use "threshold" to denote the x-axis value at the inflection point of the psychometric curve, and "slope" to denote the sensitivity or slope of the curve at this inflection point. Also all lapse rates reported, were measured through the fits of such 4-parameter logistic functions to animal's choices following previous definitions of lapse rates (Brunton et al.[30], Prins[72]) and never through the error rates at extreme stimulus strengths.

**History modulation of psychometric parameters.** To summarize the effects of trial history on psychometric parameters we fit independent psychometric curves to choices conditioned on 1-trial back choice-outcome history i.e. following rightward wins (Rw) and leftward wins (Lw). Modulation of the threshold parameter by history was then computed as $x_0^{Rw} - x_0^{Lw}$. To quantify the modulation of lapse rate parameter by history we first computed the difference in the left and right asymptotes following rightward and leftward wins: $\kappa_0^{Rw} - \kappa_0^{Lw}$ and $(\kappa_0^{Rw} + \kappa_1^{Rw}) - (\kappa_0^{Lw} + \kappa_1^{Lw})$ respectively. The net modulation of lapse rates with trial history is given by the sum of these differences: $2(\kappa_0^{Rw} - \kappa_0^{Lw}) + (\kappa_1^{Rw} - \kappa_1^{Lw})$.

## Behavioral tasks

**Auditory evidence accumulation task.** Rats were trained with previously established protocol[30,36,37,73] using the BControl system. Briefly, rats were put in an operant chamber with three nose ports. They were trained to begin a trial by poking their nose into the middle port. This initiated two simultaneous streams of randomly-timed discrete auditory clicks for a predetermined duration after a variable delay (0.5–1.3s), one from a speaker to their left and the other to their right. Rats were required to maintain "fixation" throughout the entire stimulus (1.5s), failure to do so led to a violation trial. At the end of the stimulus, rats had to poke towards the side which played the greater number of clicks to obtain a water reward. Stimulus difficulty was varied from trial-to-trial by changing the ratio of the generative Poisson rates of the two click streams. Trial difficulty and rewarded side were independently sampled on each trial.

We analyzed rats which performed greater than 30,000 trials, at 70% or more accuracy. Sessions with less than 300 trials or less than 60% accuracy for either of the choices were excluded. Since rats typically perform this task for many months after having passed the final training stage, to minimize nonstationarities in the data (due to break in training because of holiday closures etc.) and ensure that we are analyzing asymptotic performance, we identified temporally contiguous sessions with stable accuracy by performing change-point detection on smoothed trial hit rate using MATLAB's *findchangepts* function. The partition with most number of trials was included in the analysis. Since the animals neither made a choice nor received an outcome on violation trials, we ignore them while computing trial-history effects. In addition, data from 19 rats analyzed in Brunton et al.[30] was also included in this analysis.

**Auditory evidence accumulation task with reaction time reports.** To measure rats' reaction times in addition to choices we modified the auditory evidence accumulation task in two ways. First, we relaxed the "fixation" requirement and instead allowed rats to sample the stimulus for as long as they want. As soon as rats broke fixation by removing their nose from the center port, the stimulus stopped and the rats were required to report their decision by poking into one of the side ports. For any given trial, the time that the rat spent sampling the stimulus was its reaction time. Second, we rewarded rats if they correctly reported the side which had greater underlying Poisson rate rather than the side which played the greater number of clicks. This helped eliminate the trivial strategy of culminating a decision after the first click and having perfect accuracy by simply reporting the side of that click without any need for evidence accumulation.

In practice, we followed the same training protocol as the interrogation task[30] but with the modified reward rule. Once the rats were fully trained on the interrogation protocol we gradually reduced the duration of delay between stimulus onset and trial initiation as well as the fixation period. Most rats maintained high accuracy (>70%) upon this manipulation, if rats performance did not meet this criterion even after a week of training, they were excluded. Rats tended to have worse accuracy early in the session, so we omitted the first 50 trials from our analysis. After the first 50 trials, we confirmed that the accuracy in the first and second halves of the session was comparable.

## Data modeling methods

**Accumulator model.** To model subjects choices and RTs, we used the accumulation to bound model modified to take into account the discrete nature of evidence in our behavioral tasks[30]. In the model, the evolution of accumulated evidence $x(t)$ in response to the left ($\epsilon_L$) and right ($\epsilon_R$) click trains on trial $n$ is given by:

$$dx = \begin{cases} 0, & \text{if } |x| \geq B \\ \lambda x dt + (\epsilon_{R,t}C_R(t)\xi_R - \epsilon_{L,T}C_L(t)\xi_L)dt + \sigma_x dW & \text{otherwise} \end{cases} \tag{11}$$

$$\text{where } \frac{dC}{dt} = \frac{1-C}{\tau_\phi} + (\phi - 1)C(\epsilon_{R,t} + \epsilon_{L,t}) \quad \text{and} \tag{12}$$

$$x(t=0) = I(n) \tag{13}$$

where $\lambda$ is the inverse time constant of the consistent drift in memory of $x(t)$. $C_R(t)$ and $C_L(t)$ are the magnitudes of each right and left click respectively after undergoing sensory adaptation (with adaptation strength $\phi$ and adaptation time constant $\tau_\phi$). The sensory noise that accompanies each click is represented by $\xi_R, \xi_L$ which are Gaussian random variables with mean 1 and variance $\sigma_s^2$. The accumulation variable $x$ also undergoes Brownian diffusion through the addition of a Wiener process ($W$) with variance $\sigma_x^2$. $B$ represents the absorbing decision bound that prevents $x(t)$ from evolving further, if crossed. The initial value of the accumulator variable $a$ varies from trial-to-trial and is set based on exponentially filtered history of previous choices and outcomes (see Methods section on Models of initial state updating). A choice is made by comparing the final value of the accumulator $x(T)$ to a side bias. A rightward choice is made if $x(T) > $ bias.

Since the model quantifies noise sources on each trial, it requires estimating the evolution of a noise-induced probability distribution $P(x(t))$. We compute $P(x(t))$ by solving the Fokker-Planck equations that correspond to model dynamics (see refs. 30,74 for numerical methods). The probability of making a rightward choice at the end timepoint $T$ of a trial, given accumulation model parameters $\theta^{acc}$ is:

$$P(\text{choose R}|\epsilon_R, \epsilon_L, \theta^{acc}) = \int_{x=\text{bias}}^{\infty} dx P(x(T)|\epsilon_R, \epsilon_L, \theta^{acc}) \tag{14}$$

**Models of true lapses.** We assume that some fraction of choices $\kappa$ arise from processes extraneous to evidence accumulation such as motor error/exploration or inattention. We parameterize these processes with $\theta^{lapse}$ and refer to them as "true lapses":

- In the motor error/exploration variant, the probability of making a choice towards the right - when lapsing - is given by $\rho$.

$$P(\text{choose R}|\theta^{lapse}) = \rho \tag{15}$$

- In the inattention variant (Supplementary Fig. 5c), the subject lapses towards the side favored by the initial state relative to a bias $\rho$. So the probability of a rightward choice due to inattention on trial $n$ is:

$$P(\text{choose R}|\theta^{lapse}) = \begin{cases} 1 & \text{if } i(n) - \rho > 0 \\ 0.5 & \text{if } i(n) - \rho = 0 \\ 0 & \text{if } i(n) - \rho < 0 \end{cases} \tag{16}$$

- In the hybrid variant (with motor error and inattention; Supplementary Fig. 9), the probability of lapsing towards right depends on the initial state through a sigmoidal function whose

slope $m$ (or matching constant) as well as bias $\rho$ is a free parameter:

$$P(\text{choose R}|\theta^{lapse}) = \frac{1}{1 + e^{-m(i(n)-\rho)}} \tag{17}$$

Hence the total probability of making a rightward choice due to accumulation and true lapses is:

$$P(\text{choose R}|\Theta) = (1-\kappa)P(\text{choose R}|\epsilon_R, \epsilon_L, \theta^{acc}) + \kappa P(\text{choose R}|\theta^{lapse}) \tag{18}$$

where $\Theta = \{\theta^{acc}, \theta^{lapse}, \kappa\}$.

**Model fitting.** The model parameters were fit to individual rats by maximizing the log likelihood of the observed choices of the rat $\mathbf{c_{obs}}$, i.e. by maximizing

$$\ln \mathcal{L}(\mathbf{c_{obs}}|\boldsymbol{\epsilon_R}, \boldsymbol{\epsilon_L}, \Theta) = \Sigma_n \ln P(c_{obs,n}|\epsilon_{R,n}, \epsilon_{L,n}, \Theta) \tag{19}$$

where $n$ indexes trials. Throughout this manuscript, we assumed that for each rat, the parameters remain fixed across all sessions. So one set of parameters were fit to each rat for each model variant. Constrained optimization was performed in Julia using Optim package. We computed gradients for parameter optimization using a forward-mode automatic differentiation package. The reported maximum likelihood parameters and likelihood values (used for model comparison) are from model fits to the entire dataset. We fit a random subset of 10 rats using 5-fold cross-validation (85% training dataset, 15% test dataset) but this yielded very similar maximum likelihood parameters and virtually identical test and training log-likelihoods. Hence, to save on computing time we fit the different model variants to each rat's entire dataset. This agreement between test and training likelihoods is likely due to the large number of trials in our dataset and the modest number of parameters in our model.

**Simultaneous modeling of choices and RTs.** In decision-making tasks, observed reaction times (RTs) are often thought of as comprised of stimulus sampling or decision times (DTs, the time it takes for the subject's accumulated evidence to hit the bound) and non-decision related processing times (NDTs). In our datasets we observed that reaction times tended to be slower following incorrect trials and that they grew longer over the course of a session. These effects could be isolated just to RTs and were not observed in choice behavior. To model these trends we conceptualize non-decision times as arising from a separate drift diffusion process whose drift $\nu$ is additionally modulated by current trial number $n$ and previous trial's outcome. These non-decision time drift-diffusion processes terminate when the bound $\omega$ is hit. We assume that the non-decision times for each choice $k \in \{L, R\}$ have independent bounds ($\omega_k$) and drifts ($\nu_k$). So the non-decision times for a trial $n$ are samples from the following Wald or Inverse Gaussian ($IG$) distribution:

$$\tau_n^{NDT} \sim IG\left(\frac{\omega_k}{\nu_k - \alpha n + \gamma_o 1_{(n-1)}^-}, \omega_k^2\right) \tag{20}$$

where $k \in \{L, R\}$ and $1_{(n-1)}^-$ is an indicator function which is 1 if the previous trial was incorrect and is 0 otherwise. $\alpha$ parameterizes the impact of trial number on NDTs and $\gamma_o$ parameterizes the impact of previous trial's outcome on current trial's NDT.

We fit the model by maximizing the joint log likelihood of the observed choices and RTs. For any given trial, we can compute the likelihood of observing a particular reaction time $RT_{obs}$ and choice $c_{obs}$ due to accumulation by marginalizing over possible decision or bound

hitting times $\tau_{c_{obs}}$ for the observed choice:

$$P(c_{obs}, RT_{obs}|\epsilon_R, \epsilon_L, \theta^{acc}, \theta^{NDT}) = \int_0^{RT_{obs}} P(\tau_{c_{obs}}|\epsilon_R, \epsilon_L, \theta^{acc}) \atop P(c_{obs}, RT_{obs}|\theta^{NDT}, \tau_{c_{obs}}) d\tau_{c_{obs}} \tag{21}$$

On true lapse trials, RTs were assumed to arise from NDTs alone and therefore the joint likelihood due to accumulation and true lapses is given by:

$$\mathcal{L}(c_{obs}, RT_{obs}|\epsilon_R, \epsilon_L, \Theta) = (1 - \kappa) P(c_{obs}, RT_{obs}|\epsilon_R, \epsilon_L, \theta^{acc}, \theta^{NDT}) \atop + \kappa P(c_{obs}, RT_{obs}|\theta^{lapse}, \theta^{NDT}) \tag{22}$$

where $\Theta = \{\theta^{acc}, \theta^{NDT}, \theta^{lapse}, \kappa\}$.

We followed previously established methods to compute the probability distribution of $x(t)$ for computing the likelihood[30,74]. This involves expressing the temporal dynamics of the probability distribution as a Fokker-Planck equation and then computing the solution numerically, by dividing $P(x(t))$ into a set of n discrete spatial bins and determining how probability mass moves after a discrete temporal interval $\Delta t$. The transition matrix for discrete time dynamics and a full description of the methods can be found in these studies.

### Reporting summary

Further information on research design is available in the Nature Portfolio Reporting Summary linked to this article.

## Data availability

The rodent behavioral data generated in this study from the Poisson Clicks and the reaction time task have been deposited in the figshare database under accession code CC BY 4.0 at the following https://doi.org/10.6084/m9.figshare.24113793. Source data are provided with this paper.

## Code availability

Analysis codes is available here: https://github.com/Brody-Lab/trialhistory_lapses_EA.git with https://doi.org/10.5281/zenodo.10161051.

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

## Acknowledgements

We thank members of the Brody lab for experimental support and helpful feedback throughout the project especially Adrian Bondy, Thomas Luo, Emily Dennis, Tyler Boyd-Meredith, and Ahmed El-Hady. We also thank Jovanna Teran and Brody lab technicians for assistance with rat training. We are grateful to Sashank Pisupati, Jonathan Cohen, Sebastian Musslick, Jonathan Pillow, and Ilana Witten for helpful discussions at various points during the project. This work was supported by NIH grant R01MH108358 awarded to C.D.B as well as a grant from the Simons Foundation (Grant number: NC-GB-CULM-00003118-03) awarded to C.D.B.

## Author contributions

D.G. organized and analyzed the data and wrote the initial draft of the manuscript. C.D.K. assisted with data collection. B.D. assisted with analysis. All authors provided feedback on the manuscript. C.D.B. oversaw all aspects of the project.

## Competing interests

The authors declare no competing interests.
