## [Peer Review File · Nature Communications]

Trial-history biases in evidence accumulation can give rise to apparent lapses in decision-makingREVIEWER COMMENTS

Reviewer #1 (Remarks to the Author):

The manuscript by Gupta et al., has examined how lapses arise during decision-making. Unlike most previous studies that have suggested distinct processes underlie lapses vs. trial history, this study concludes that both suboptimalities arise from the same cognitive process that is due to the world being non-stationary. The authors propose that both suboptimalities can be explained by a trial-history dependent update to the initial state value in the Drift Diffusion Model. They show that their model does a better prediction of rats behavior in a large dataset including distinct decision making tasks.

The manuscript addresses an interesting question in the field and is well-written. However, for it to be published at Nature Communications with a broad audience in all fields of neuroscience, particularly systems neuroscience, the paper should address the following points:

1. While the model attributes trial history and lapses to small and large magnitudes of the initial state value, respectively, and hence, concludes that both suboptimalities arise from the same process, they may still correspond to distinct neuronal mechanisms. Discussing this topic and the potential neural pathways and sites that may underlie different values of the initial state will be helpful.
2. While the model performs well on the current dataset, it is unclear how lapses may arise in the following cases: 1) task designs that actively combat trial history bias but in which lapses can still occur due to the animal's internal state variations, e.g., exploration, inattention, etc.; 2) data where incorrect trials have longer reaction times? Please consider discussing this topic.
3. A non-stationary world may lead to trial history biases (hence lapses as suggested by this work) due to dynamic updating of other DDM parameters, e.g., change in decision bound, or change in impulsiveness, etc. Why did the authors not consider these alternatives, and only focused on the initial state value as a potential mechanism underlying trial history biases? Is it likely for alternative models that explain trial history biases by these other parameters to perform better than the current model? This should be at least discussed in the paper, otherwise, attributing these suboptimalities to the initial state seems non-conclusive in the absence of comparison of the performance of the current model with alternative models.

Minor points:

Line 11: “normative under misbeliefs about non-stationarity in the world”:

Please consider rephrasing to make the abstract easily understandable by a larger community, outside computational neuroscience.

Line 63: typo: “the a”: typo.

Line 379: typo: it should be "capture".

Reviewer #2 (Remarks to the Author):

The authors tackle the well studied problem of lapses and biases in perceptual decision making. It has been shown before that lapses and biases may be linked. Moreover, theoretical models indicate that apparent lapses could be due to history dependent biases in initial beliefs due to an animal's mistaken assumption about the dependence of trials. Models that include such history dependence make several concrete predictions about lapses, and the timing of responses on particular sequences of trials. The authors convincingly show that data from a large number of trials with rats agree with the predictions of these models. This indicates that at least some lapses may be a consequence of a wrong model about the world, rather than inattention or motor noise.

In the case that the change rate is known (ie the rate of jumps between the two options), then the initial value will depend only on the previous outcome (win or loss), and not a filter. This is just a consequence of the Markovian assumption in the analysis in eg. Yu & Cohen 2009, Nguyen et al 2019. However, it also gives a more parsimonious model that the current model should be compared to. Moreover, I believe that this model leads to the same predictions.

I would suggest cross-validation rather than BIC when comparing different models to data (eg Fig. 3D). BIC has well known issues. I understand that may be tricky due to the true lapses. However, if true lapses are really low, then if the authors' claim that most of the decisions are driven by a deterministic strategy should lead to good predictability on individual trials. If this is not the case, then the authors should explain why.

This could also help the fact that a model that does a worse job in fitting the data is chosen when using the BIC (line 301). Right now this observation needs to be explained away, which raises questions about the usefulness of the BIC in this context.

Also, some clarification

- footnote on p. 3 “time-varying bounds on the posterior” - I assume the bound is on the accumulated evidence. I am not sure what a bound on the posterior is.

- I am not sure what the exact definition of lapse rate and apparent lapses: Does the lapse rate necessarily have to be the same at both asymptotes? This is a consequence of the symmetry assumed in across trial processing, but should be explained. Also, what exactly is an “apparent lapse”? Does any trial on which the initial bias points in the wrong direction, that also results in a wrong choice count? If not, at what level of initial bias do we talk about a lapse?

The issue may be that the authors are talking about the average effect over many trials, which makes the definition of an “apparent lapse” on a single trial a bit confusing.

- The new task described on p. 14: What is the difference between detecting a higher number of clicks and a higher Poisson rate? I think in a fixed interval the two would be the same. Is there a larger difference in the case of a free responses?

The caption in Fig. 4A says that the evidence is the logarithm of the ratio of Poisson rates. However, the rat does not have direct access to the rates, only the click counts. This is confusing.

Check for typos, eg

l. 56, raises -> raise

l. 63 “ that the a substantial”

These are just some examples. There are not that many, and they can be found with a careful reading or two.

Reviewer #3 (Remarks to the Author):

Gupta et al. present a rigorous study of the role of recent experience in shaping sensory decision-making. Two pervasive features of decision-making across humans, non-human primates, and rodents are that 1) our recent choices and their outcomes impact current choices even when they are irrelevant to the task and 2) that we sometimes seem to ignore the evidence in front of us and choose randomly (lapses). These are typically characterized as independent features of psychometric functions. In a very large data set of rat behavior, the authors show that both these effects can arise from a single mechanism, which is the impact of past choices on the initial state of an evidence accumulation process that generates perceptual decisions. They argue that these evidence history effects can be interpreted as a form of statistical inference about the task environment under the (incorrect, but reasonable) belief that the task environment changes over time.

I believe this paper is of general interest to cognitive psychologists and neuroscientists. Perceptual decision-making tasks are a ubiquitous tool for probing the nervous system and psychological constructs, but behavior in these tasks is quite subtle and varied. Characteristics of behavior such as choice autocorrelation and lapses are often treated as nuisances because they do not neatly fit into existing models of decision-making, yet the reality is that such nuisances often have as much or more explanatory power as common experimental manipulations. This paper parsimoniously integrates lapses into existing models of behavior, which, in addition to being interesting for its own sake, will give other researchers tools to better understand how their own manipulations of interest impact the brain and behavior.

The study is thorough and convincing. The authors do a very good job of explaining the link between their mathematical models and their substantive scientific question, which is a place modeling-heavy papers often struggle. They also provide ample evidence that their model captures the relevant empirical regularities in behavioral data. There are, however, a few spots where I feel that the evidence presented does not quite reach the claims they make about it.

Substantive concerns

1. I have some concern that the fitted models of trial history effects may not support the scientific interpretation the authors present. Specifically, the authors link the trial history effects to a process of

statistical inference about a constantly changing world. However, the functional form of the trial history effects the authors used is very flexible (equation 1), and a flexible nonlinear model such as this can exhibit all sorts of unexpected behaviors and capture regularities in animal behavior that are unrelated to the intended interpretation of the model. The authors point out that certain parameter values of their model correspond to interpretable forms of statistical inference, but we are not shown that this holds in the rats. Are the choice history effects estimated from the rats consistent with any recognizable model of statistical inference? Or are they interpretable in any other way? At a minimum, the authors should show the distribution of the estimated parameters of equations (1) and (2) across rats so that readers can judge for themselves. I recognize that the authors have attempted to clarify the difference between their empirical models and their interpretations (e.g. in ll. 350-356), but more of this up front in the paper might help.

2. The second issue relates to the models of “true lapses”. My intuition is that the deterministic nature of the “inattention variant” would render it very poorly suited to modeling behavioral data, and that the “hybrid variant” is a much stronger and more practical candidate model. However, the hybrid model is fit only to the RT data set, even though as far as I can tell it is just as applicable to the full sample. I would much prefer to see either both models, or just the hybrid model used to analyze the full sample in Supp. Figure 4.

3. Noting that augmenting H1St with the “inattention variant” of true lapses does not change the model fits much, the authors claim on page 14 that “apparent lapses produced by history-dependent initial states (rather than true lapses due to motor error or inattention) are the major driver of history-dependent co-modulations in psychometric thresholds and lapse rates in the dataset.” I do not think this claim is justified by the results they present. If adding a new feature to a model does not change its predictions, this does not mean that we can conclude that this feature is not important. It could also be that the new feature is redundant with previous features of the model. For example, in a linear regression, adding a new covariate that is highly correlated with the previous covariates will not change the model fit, but this does not mean that the new covariate is less important than the previous covariates; instead, it means we cannot distinguish between them.

In this paper, it could be that choice-history effects on the accumulation process are indistinguishable from choice-history effects on inattention-driven lapses. To demonstrate that this is not the case, I would suggest fitting a model where choice history can only affect lapses. Under the author’s hypothesis, this model should be much worse than any model that allows choice history to impact the accumulation process. Again, I would prefer to see this comparison done with the hybrid lapse model as well as (or just instead of) the inattention variant.

4. Finally, it would be interesting to know what proportion of lapses are “apparent” versus “true” in the rodents’ choices. The “true” lapse rate is encoded directly in the parameter κ , while the “apparent”

lapse rate could presumably be extracted from the implied psychometric curves of the accumulator model.

Minor issues and clarifications:

- Are the parameters for each rat assumed fixed across sessions? It seems so but this should be explicitly noted in the methods section.

- Pg 10, 188, "Across rats, 17% of total lapse rates are modulated by trial history", and Supp. Fig 3D: I have no idea what analysis was done here. How many lapse rates are calculated for each rat, and how? Apologies if I am missing this through my own inattention.

- Supp Fig 2A, "Blue and pink curves are conditioned on past left/right wins". All past left/right wins, or only the last trial being a left/right win? In behavior plots, it seems to be conditioned on the last trial, but here it is ambiguous. Also, please give explicit parameter values instead (or in addition to) high/low, etc.

- Fig 3D and Supp Fig 4E are much less informative than they could be. I recommend showing the distribution of BIC differences across rats. Apply this to any other max BIC plots I have forgotten.

- Page 13, 227-230, "Across individuals, the model with HIST captured...": I do not understand what analysis was actually done here. It must refer to the results displayed in Fig 3F & G, but this is not stated in the text. The description of the analysis in the figure caption is more clear, but still needs elaboration. How are history modulations of thresholds and lapse rates quantified from the HIST model?

- Early in the methods (e.g., bottom of p21), x is used as the accumulated evidence variable and ϵ for the increments. However, starting on p25, the authors use the notation of Brunton et al., where accumulated evidence is $a(t)$, clicks are δ , and the definition of η is different than the one used in the filtering model for trial history effects. It would be helpful if the authors harmonize their notation here.

- The joint model of choices and RTs (page 27-28) contains insufficient detail on the fitting process. The integral in the penultimate equation presumably does not have a closed form solution, so how was the model estimated?

Reviewer #4 (Remarks to the Author):

This study aims to provide a more unified view of certain common suboptimalities in decision-making behaviors, typically described as biases and lapses, in terms of mistaken beliefs about the stationarity of the task environment that drive history-dependent adjustments in behavior. They develop a computational model based on these ideas and show that it does a nice job fitting rodent data.

I think the basic ideas are reasonable, interesting, and valuable to the field (that can always use more precise and accurate models of behavior); the model is appropriate and provides a clear, quantitative description of the data; and the paper is well written. All that said, I also have several major concerns, detailed below, that should be addressed:

1. Much of the framing makes the basic ideas sound more novel than I think they are. I kept wanting a simpler description of the phenomenon and the model, something along the lines of, “we’ve known since at least Laming (1968) that decision-makers often show suboptimal sequential biases, and we’ve known from countless studies that biases can have several well-known effects on psychometric (and chronometric) functions, so if you put those together and the sequential biases are big enough you can get these effects.” Again, I think putting it together is useful, but from a theoretical perspective the ideas aren’t new, and I think better framing them in terms of what is known will make them more accessible. Some related points:

a. The technically most interesting/relevant part of the story is exactly how a (true or apparent) lapse is measured. As the authors note, a lapse is considered an “asymptotic” error – but exactly how the asymptote is defined and measured can vary widely across studies. Of course, apparently small differences in where/how the asymptote is measured can have a major role in how susceptible the measurement is to the kind of biasing influence they posit (e.g., consider an initial bias of something like 0.731, corresponding to log-odds of ~ 1 ; if the “asymptote” is measured for “strong” evidence corresponding to log-odds of 3, then the bias could provide a very measurable change from $\sim 95\%$ choices to $\sim 88\%$ choices; however, if stronger evidence corresponding to a log-odds of ~ 6 is used, then the effects of the biasing term might be negligible, going from 99.8% to 99.3% choices). It might be useful to provide some insights into how these considerations relate to the conditions in which one might or might not expect to see the effects reported.

b. Relatedly, it would be useful to provide more intuition for why/when the model gives predictions that seem to correspond to either a flattened asymptote on the psychometric curve that traditionally would be thought of as a “lapse” (e.g., Fig. 3B, right panel) or curves that do not seem to have yet reached an asymptote and thus would not traditionally be thought of as being useful for providing a good estimate of lapses (e.g., Fig. 3C, right panel).

c. It also seems like it could be useful to be clearer in places about the definition of “lapses.” For example, the Discussion states that “Lapse rates are often considered to be a mixed bag comprising several different noise processes, yet most studies so far have focused on one or more of these

component processes in isolation.” This seems vague to me and misses the key point that the traditional definition of lapses emphasizes that they are based on factors that are entirely separate from the decision process (e.g., “finger slips”).

d. As noted above, the main behavioral effect driving the results seems to be that the rats have pretty big, suboptimal sequential biases. Seems like it would be useful to discuss that finding (and the degree to which is likely does and does not support generalization of the findings to other species/tasks/conditions) in more detail.

2. It also would be nice to provide some more background about what is and is not new about the basic model. For example, Kim et al (J Neurosci, 2017) also used an “accumulation to-bound model with history-dependent updates to the initial state of the accumulator” where the updates were based on “Bayesian learning of priors under the belief that the prior probabilities of the two hypotheses can undergo un signaled jumps.”

Minor comments:

1. I found the description of Eq. 3 (Lines 110–121) to be a bit confusing, in several ways: a) saying “small deviations in the initial state largely manifest as additive biases to the evidence, shifting psychometric curves horizontally towards the option favored by the initial state” seems like an odd way of describing an effect that is most definitely not an additive bias to the momentary evidence (which can of course be modeled explicitly) but rather a change in total evidence (maybe instead of “largely manifest as” you can say “resemble” if you really want to make this point); and b) calling a shift in this kind of psychometric function (where choice is on the y axis) as affecting “threshold” also may be a bit confusing, since “threshold” is usually a measure of sensitivity, which as described a few lines below is reflected in the slope, not the shift, of these curves.

REVIEWER COMMENTS

Color key:

(Reviewer comments)

(Response to reviewers)

(excerpts from manuscript)

(Text in bold is newly added excerpts from manuscript)

Reviewer #1 (Remarks to the Author):

The manuscript by Gupta et al., has examined how lapses arise during decision-making. Unlike most previous studies that have suggested distinct processes underlie lapses vs. trial history, this study concludes that both suboptimalities arise from the same cognitive process that is due to the world being non-stationary. The authors propose that both suboptimalities can be explained by a trial-history dependent update to the initial state value in the Drift Diffusion Model. They show that their model does a better prediction of rats behavior in a large dataset including distinct decision making tasks.

The manuscript addresses an interesting question in the field and is well-written. However, for it to be published at Nature Communications with a broad audience in all fields of neuroscience, particularly systems neuroscience, the paper should address the following points:

We thank the reviewer for their kind words, and have prepared a point-by-point response.

1. While the model attributes trial history and lapses to small and large magnitudes of the initial state value, respectively, and hence, concludes that both suboptimalities arise from the same process, they may still correspond to distinct neuronal mechanisms. Discussing this topic and the potential neural pathways and sites that may underlie different values of the initial state will be helpful.

We agree with the reviewer that these issues should be discussed, and trial history effects and lapses may indeed arise through disparate mechanisms; our manuscript focuses on a potential mechanism that jointly gives rise to both. We have therefore extended the discussion of potential pathways, and we have redoubled efforts to make sure that we describe the joint source view as a possibility, supported by our model and analysis, not as a certainty.

We have amended our discussion to emphasize these points:

Line 443:

Our model predicts that an increased reliance on history (i.e., larger shifts of the initial states) should produce more apparent lapses. Indeed, this could provide an explanation that links disparate sets of observations from previous studies: while some studies have reported that perturbations of secondary motor cortex and striatum give rise to higher lapse rates (Erlich, Brunton, et al., 2015; Yartsev et al., 2018; Guo et al., 2019; Sindreu et al., 2021; Pisupati et al., 2021), others have shown that the effects of perturbing these regions seems to resemble an increased history-dependence (Sindreu et al., 2021; Luo et al., 2021). Interpreting these results

through the lens of our model, we would conclude that these regions play a crucial role in the interaction of history-dependent initial states with sensory evidence, **making them a potential common neural substrate that could contribute to both kinds of suboptimalities**. Indeed, increased history dependence upon c4M2 perturbation has been shown to be mediated by increased bias in initial value of the neurally derived accumulator variable (Luo et al., 2021). **Similarly, DMS perturbations had large effects on lapse rates in moderately engaged behavioral states that were influenced by both sensory evidence and history (Bolkan et al., 2022)**. Our model could also help explain why Busse et al. 2011 found that mice with higher lapse probabilities showed higher history dependence, or results from The International Brain Laboratory et al. 2021 who observed a modulation in lapse rates in addition to horizontal biases upon explicit manipulation of category priors. **Nonetheless, these observations do not preclude the possibility that there are indeed independent neural mechanisms and/or areas through which trial-history effects and lapses (particularly true lapses) arise. Consistent with this, studies have implicated different brain areas in producing deterministic vs stochastic biases in action timing (Murakami et al., 2017), and even different sub-circuits within the same area in giving rise to distinct behavioral strategies (Cazettes et al., 2021). Detailed manipulations of brain regions with prior information such as in studies like (Findling et al., 2023) could help pinpoint the neural mechanisms through which these suboptimalities arise.**

(Text in bold is newly added)

2. While the model performs well on the current dataset, it is unclear how lapses may arise in the following cases: 1) task designs that actively combat trial history bias but in which lapses can still occur due to the animal's internal state variations, e.g., exploration, inattention, etc.; 2) data where incorrect trials have longer reaction times? Please consider discussing this topic.

We agree with the reviewer that there are other potential sources of lapses (motor error, inattention, exploration etc) that arise independent of initial state updating, we denote such processes as “true lapses” distinct from “apparent lapses” that arise from initial state updating - in fact we accounted for these “true lapses” in all our behavioral models and we expect them to capture lapse rates in scenarios where trial history bias is minimized or actively combated. In datasets, with longer reaction times, these other sources as well variability in drift-rates could give rise to lapses.

We have extended our discussion to highlight these points:

Line 416:

Another crucial possibility, trial-to-trial variability in drift rates - is known to give rise to longer error RTs than correct RTs (Ditterich, 2006a; Ditterich, 2006b; Drugowitsch, Moreno-Bote, et al., 2012; Nguyen and Reinagel, 2020), which is a signature often reported in monkeys and humans (Roitman and Shadlen, 2002; Shevinsky and Reinagel, 2019). **We did not observe the reaction time signatures of drift rate variability in our dataset, instead we identified signatures of initial state variability, where error RTs were shorter than correct RTs, rather than longer. However, drift rate updates may represent an alternative mechanism through which history-modulated apparent lapses could occur in other datasets. It is worth noting that certain task designs include efforts to actively measure and counter trial history biases. In such cases, lapses may still occur, likely due to exploration or inattention. In this**

manuscript, we refer to lapses caused by these factors “true lapses”, as they cannot be explained by fluctuations in DDM-related parameters.

3. A non-stationary world may lead to trial history biases (hence lapses as suggested by this work) due to dynamic updating of other DDM parameters, e.g., change in decision bound, or change in impulsiveness, etc. Why did the authors not consider these alternatives, and only focused on the initial state value as a potential mechanism underlying trial history biases? Is it likely for alternative models that explain trial history biases by these other parameters to perform better than the current model? This should be at least discussed in the paper, otherwise, attributing these suboptimalities to the initial state seems non-conclusive in the absence of comparison of the performance of the current model with alternative models.

We thank the reviewer for raising this important point. Indeed, a non-stationary world may warrant updates in a number of key decision variables: priors, likelihoods, reward functions and their equivalent DDM parameters. However, in this manuscript we exclusively considered initial state updates for the following reasons:

1. Normative updates in all of these variables impact initial states, but certain updates also influence drift rate (or decision bounds in an alternative parametrization). This led us to primarily focus on the dynamics of initial state updating.
2. Initial state updating was able to explain an impressive amount of variance in our dataset - hence we claim that at the very least, initial state updates are a major contributor to this effect.
3. Given that our dataset shows strong signatures of initial state variability (e.g. shorter error RTs) and not of drift rate/decision bound variability (longer error RTs), we think it is quite unlikely that drift rate variability is a major player in our dataset.

We have made modifications to our discussion to provide a more comprehensive explanation of our rationale for this decision and clarified the extent of our conclusions:

Line 399:

In our treatment, we only considered history-dependent updates to the initial state of a DDM. Such a mechanism is normative under non-stationary beliefs about the prior c_1 , which is our favored interpretation as it aligns with other studies of history biases (Gold, Law, et al., 2008; Yu and Cohen, 2009; Summerfield and Koechlin, 2010; Goldfarb et al., 2012; Mulder et al., 2012; Abrahamyan et al., 2016; Kim et al., 2017; Molano-Mazon et al., 2021). Nevertheless, these updates may also reflect other heuristic strategies (Gigerenzer and Gaissmaier, 2011) which we accommodate using our flexible parameterization of initial state updates. Animals may entertain non-stationary beliefs about other elements of the decision process, such as the rewards or likelihoods (Dayan and Daw, 2008; Mendonça et al., 2020; Lak et al., 2020; Pisupati et al., 2021). Normative updating in such situations still reduces to initial state updates in simple settings (for e.g. non-stationary rewards for a single difficulty; Simen et al. 2009; Rorie et al. 2010), but in more complex ones it **affects drift rates or bounds in addition to initial states** (Palmer et al., 2005; Eckhoff et al., 2008; Hanks, Mazurek, et al., 2011; Drugowitsch, Mendonca, et al., 2019; Fan et al., 2018; Urai et al., 2019; Mendonça et al., 2020). **This commonality of initial state updating to many different non-stationary beliefs motivated us to probe its role in producing apparent lapses, and indeed this mechanism was able to explain an impressive amount of variance in our dataset, leading us to conclude that initial state updating is at least a major factor driving animal behavior. Another crucial possibility,** trial-to-trial

variability in drift rates - is known to give rise to longer error RTs than correct RTs (Ditterich, 2006a; Ditterich, 2006b; Drugowitsch, Moreno-Bote, et al., 2012; Nguyen and Reinagel, 2020), which is a signature often reported in monkeys and humans (Roitman and Shadlen, 2002; Shevinsky and Reinagel, 2019). **We did not observe the reaction time signatures of drift rate variability in our dataset, instead we identified signatures of initial state variability, where error RTs were shorter than correct RTs, rather than longer. However, drift rate updates may represent an alternative mechanism through which history-modulated apparent lapses could occur in other datasets.**

Minor points:

Line 11: “normative under misbeliefs about non-stationarity in the world”:

Please consider rephrasing to make the abstract easily understandable by a larger community, outside computational neuroscience.

We have rephrased this as: “optimal under mistaken beliefs that the world is changing i.e. nonstationary”:

Line 10:

Here we demonstrate that history biases and apparent lapses can both arise from a common cognitive process that is **optimal under mistaken beliefs that the world is changing i.e. nonstationary**.

Line 63: typo: “the a”: typo.

Fixed

Line 379: typo: it should be "capture".

Fixed

Reviewer #2 (Remarks to the Author):

The authors tackle the well studied problem of lapses and biases in perceptual decision making. It has been shown before that lapses and biases may be linked. Moreover, theoretical models indicate that apparent lapses could be due to history dependent biases in initial beliefs due to an animal's mistaken assumption about the dependence of trials. Models that include such history dependence make several concrete predictions about lapses, and the timing of responses on particular sequences of trials. The authors convincingly show that data from a large number of trials with rats agree with the predictions of these models. This indicates that at least some lapses may be a consequence of a wrong model about the world, rather than inattention or motor noise.

We thank the reviewer for these kind words.

In the case that the change rate is known (ie the rate of jumps between the two options), then the initial value will depend only on the previous outcome (win or loss), and not a filter. This is just a consequence of the Markovian assumption in the analysis in eg. Yu & Cohen 2009, Nguyen et al 2019. However, it

also gives a more parsimonious model that the current model should be compared to. Moreover, I believe that this model leads to the same predictions.

We thank the reviewer for this suggestion. In our manuscript, we had already demonstrated that our history filter effectively captures updates from Yu & Cohen 2009 (figure duplicated below) - we have now included a comparison of BIC scores from fits of both models to the dataset, and found the parsimonious model to be insufficient.

Discussion of Yu & Cohen (figure from the original manuscript)

As the reviewer correctly points out, in the case of a markovian prior (Dynamic Belief Model or DBM) the recursive form of the optimal initial state update rule only depends on the previous outcome - although it is worth noting that in its non-recursive form, the DBM's initial state is still a linear exponential filter over all past outcomes (Yu & Cohen 2009).

The DBM update rule corresponds to a restricted parameter regime of our history filter, where β^h (decay factor) and η^h (increment factor) are the same for all h where h represents the possible choice-outcome pairs (Methods: Models of initial state updating), and we show that this restricted regime accurately approximated the optimal DBM update rule (Supplementary Fig 1, copied below).

Supplementary Figure 1: Exponential filtering for initial state setting approximates Bayesian prior updates under assumptions of non-stationarity **A:** Example of a mis-belief in a non-stationary prior. Traces represent belief about prior probability of two hypotheses H^1 and H^2 inferred from a random sequence of trials drawn from a stationary symmetric prior, under the misbelief that the prior is occasionally undergoing unsignalled jumps. Such an assumed generative model is often referred to as the Dynamic Belief Model (DBM; Yu and Cohen 2009). **B:** Initial state updates corresponding exactly to the fluctuating prior beliefs in (A) that emerge from Bayesian learning (black line), plotted against approximate initial states derived from exponential filtering (dotted red line) of past choices and outcomes. The exponential filter provides a good approximation of exact Bayesian updates, while being more expressive and flexible to capture the possibility of other generative models and corresponding update rules.

New panels added to supplementary figure 7: We have now included a comparison of this reduced version of the model (11 parameters, compared to 14 in the full model) with our history filter and found it to be worse at capturing the rodent data, hence we believe that this simplified model does not fully capture the animal's nonstationary beliefs.

(B) Comparison of the “HIST” model with a reduced version that approximates optimal statistical inference in the Dynamic Belief Model (Supp Fig 1, Yu and Cohen 2009). The optimal update rule for the DBM corresponds to a restricted regime of the HIST model, where the magnitude and timescale parameters are set to be the same for all trial types i.e. left-win, right-win, left-loss and right-loss. We compared the fits of this “Reduced HIST” model with the HIST model, and found that it was not supported by BIC in the majority of rats (Left panel: overall bar height denotes the total number of rats for which that model variant scored the lowest BIC score, HIST was winning model in 105/152 rats). Moreover, the unconstrained HIST model performed better on average per trial (Right panel: Model comparison using BIC by pooling per trial BIC score across rats and computing mean. Lower scores indicate better fits. Mean per trial BIC scores across rats were significantly lower for model with HIST, $p = 2.23 \times 10^{-6}$, paired t-test. Error bars are SEM). Overall, this means that while some rats’ (47/152) updating strategy was consistent with optimal inference in a DBM, most rats’ parameters did not occupy this restricted regime. (C) Joint distribution of updating

I would suggest cross-validation rather than BIC when comparing different models to data (eg Fig. 3D). BIC has well known issues. I understand that may be tricky due to the true lapses. However, if true lapses are really low, then if the authors’ claim that most of the decisions are driven by a deterministic strategy should lead to good predictability on individual trials. If this is not the case, then the authors should explain why.

This could also help the fact that a model that does a worse job in fitting the data is chosen when using the BIC (line 301). Right now this observation needs to be explained away, which raises questions about the usefulness of the BIC in this context.

We thank the reviewer for this suggestion. We eschewed n-fold cross validation due to the computational burden (8 hours per rat per model variant x 152 rats x 2 models x n folds ~100n days), however we managed to perform a single fold of cross validation across all rats with an 80-20 split, and verified that it produces qualitatively similar results to BIC in terms of model comparison. These results are included in the manuscript as part of Supplementary Figure 6.

We have also clarified our interpretation of the results in line 301 - our primary intention here was to assess the contribution of history-dependent initial states (HlSt) to the joint modulation of thresholds and lapse rates, rather than about the overall best fitting model per se (which we nevertheless report).

Edited text (Line 325):

We also fit a hybrid variant of the accumulator model with HlSt that flexibly allows true lapses to be motor-error like and unaffected by history, or inattention-like and additionally be modulated by history (Supp Fig. 8A,B). While this model has a better BIC and leads to a slight improvement in correspondence to the history modulation of psychometric lapse rates, it does so at the cost of correspondence to modulations in psychometric thresholds (Supp Fig. 8C-E). **This equivocal improvement over the HlSt model in capturing the threshold and lapse rate modulations support the conclusion that HlSt and its resultant apparent lapses (rather than true lapses) are a major contributor to the observed co-modulation of both parameters.**

Also, some clarification

- footnote on p. 3 “time-varying bounds on the posterior” - I assume the bound is on the accumulated evidence. I am not sure what a bound on the posterior is.

We apologize for this confusion, we are referring to the posterior probability i.e. the net probability in favor of hypothesis 1, this includes the contribution of likelihood (from sensory observations) as well as prior probability. The ratio of log of these posterior probabilities, is the accumulator variable (as in Fig 1B). We have made this explicit in the manuscript:

Page 3, footnote:

In tasks where the reliability of incoming evidence (controlled by stimulus strength) varies from one trial to the next, it has been shown that ideal observers should have **time-varying bounds on the net posterior probability (combination of likelihood and prior)** (Drugowitsch, Moreno-Bote, et al. 2012).

- I am not sure what the exact definition of lapse rate and apparent lapses: Does the lapse rate necessarily have to be the same at both asymptotes? This is a consequence of the symmetry assumed in across trial processing, but should be explained.

We do not assume any symmetry in lapse rates, the sigmoidal fits used to measure lapse rates (equations copied below from Methods) have two parameters to separately account for the two asymptotes. Further, no symmetry is assumed in our models of true or apparent lapses, which both have parameters that can account for asymmetries.

Psychometric curves Psychometric curves model the probability of a subject choosing one of the options (e.g. right) as a function of stimulus strength. We parametrize the psychometric curve as a 4-parameter logistic function:

$$P(\text{choose Right}) = \kappa_0 + \frac{\kappa_1}{1 + e^{-b(x-x_0)}}$$

432 where x_0 is the threshold parameter that additively biases the stimulus x , b measures sensitivity to
433 the stimulus, κ_0 is the left asymptote or left lapse rate and κ_1 scales the logistic function. Therefore,
434 the right asymptote is given by $\kappa_0 + \kappa_1$ and the right lapse rate itself is given by $1 - (\kappa_0 + \kappa_1)$.

Also, what exactly is an “apparent lapse”? Does any trial on which the initial bias points in the wrong direction, that also results in a wrong choice count? If not, at what level of initial bias do we talk about a lapse?

The issue may be that the authors are talking about the average effect over many trials, which makes the definition of an “apparent lapse” on a single trial a bit confusing.

We agree with the reviewer that apparent lapses emerge when averaging across many trials (Fig 1E, Results: pg 3-8) i.e. the average choice behavior obtained by pooling choices with different initial state biases yields a heavy-tailed psychometric curve that appears to have asymptotic errors (lapse rates).

We have clarified this in our definition of apparent lapses:

Line 155, Pg 8 (Box):

Some definitions:

- Lapse rate: Lapse rates capture the difference between perfect performance and observed performance at the asymptotes, measured through sigmoidal fits to the psychometric curves.
- True lapse: A true lapse is an independent cognitive process through which agents generate stochastic evidence-independent choices.
- Apparent lapses: Apparent lapses are deterministic evidence-dependent choices that nonetheless contribute to lapse rates **when performance is averaged across trials.**”

- The new task described on p. 14: What is the difference between detecting a higher number of clicks and a higher Poisson rate? I think in a fixed interval the two would be the same. Is there a larger difference in the case of free responses?

We agree with the reviewer that the *inferred* Poisson rate and a higher number of clicks are equivalent in a fixed interval setting (however rewarding rats for the true higher Poisson rate vs rewarding them for the true higher number of clicks still differs especially on hard trials when there is a small difference in Poisson rates between the two sides)

However as we point out in the methods (copied below) in a free response setting, rewarding the rats for choosing a higher number of experienced clicks (whenever they leave) has the downside of encouraging a trivial strategy of culminating a decision after the first click and having perfect accuracy by simply reporting the side of that click without any need for evidence accumulation. In order to eliminate this strategy and encourage evidence accumulation, we rewarded them for the true poisson rate instead.

Line 549:

...we rewarded rats if they correctly reported the side which had greater underlying Poisson rate rather than the side which played the greater number of clicks. This helped eliminate the trivial strategy of culminating a decision after the first click and having perfect accuracy by simply reporting the side of that click without any need for evidence accumulation.

The caption in Fig. 4A says that the evidence is the logarithm of the ratio of Poisson rates. However, the rat does not have direct access to the rates, only the click counts. This is confusing.

We follow the convention from previous studies (e.g. Brunton '13, Odoemene '18), of always plotting psychometric curves as a function of the *true rewarded feature*, so that a clear category boundary can be drawn along the x-axis.

Check for typos, eg

l. 56, raises -> raise

Fixed

l. 63 " that the a substantial"

Fixed

These are just some examples. There are not that many, and they can be found with a careful reading or two.

Reviewer #3 (Remarks to the Author):

Gupta et al. present a rigorous study of the role of recent experience in shaping sensory decision-making. Two pervasive features of decision-making across humans, non-human primates, and rodents are that 1) our recent choices and their outcomes impact current choices even when they are irrelevant to the task and 2) that we sometimes seem to ignore the evidence in front of us and choose randomly (lapses). These are typically characterized as independent features of psychometric functions. In a very large data set of rat behavior, the authors show that both these effects can arise from a single mechanism, which is the impact of past choices on the initial state of an evidence accumulation process that generates perceptual decisions. They argue that these evidence history effects can be interpreted as a form of statistical inference about the task environment under the (incorrect, but reasonable) belief that the task environment changes over time.

I believe this paper is of general interest to cognitive psychologists and neuroscientists. Perceptual decision-making tasks are a ubiquitous tool for probing the nervous system and psychological constructs, but behavior in these tasks is quite subtle and varied. Characteristics of behavior such as choice

autocorrelation and lapses are often treated as nuisances because they do not neatly fit into existing models of decision-making, yet the reality is that such nuisances often have as much or more explanatory power as common experimental manipulations. This paper parsimoniously integrates lapses into existing models of behavior, which, in addition to being interesting for its own sake, will give other researchers tools to better understand how their own manipulations of interest impact the brain and behavior.

The study is thorough and convincing. The authors do a very good job of explaining the link between their mathematical models and their substantive scientific question, which is a place modeling-heavy papers often struggle. They also provide ample evidence that their model captures the relevant empirical regularities in behavioral data. There are, however, a few spots where I feel that the evidence presented does not quite reach the claims they make about it.

We thank the reviewer for their kind words, and have prepared a point-by-point response.

Substantive concerns

1. I have some concern that the fitted models of trial history effects may not support the scientific interpretation the authors present. Specifically, the authors link the trial history effects to a process of statistical inference about a constantly changing world. However, the functional form of the trial history effects the authors used is very flexible (equation 1), and a flexible nonlinear model such as this can exhibit all sorts of unexpected behaviors and capture regularities in animal behavior that are unrelated to the intended interpretation of the model. The authors point out that certain parameter values of their model correspond to interpretable forms of statistical inference, but we are not shown that this holds in the rats. Are the choice history effects estimated from the rats consistent with any recognizable model of statistical inference? Or are they interpretable in any other way? At a minimum, the authors should show the distribution of the estimated parameters of equations (1) and (2) across rats so that readers can judge for themselves. I recognize that the authors have attempted to clarify the difference between their empirical models and their interpretations (e.g. in ll. 350-356), but more of this up front in the paper might help.

Thank you for this suggestion, we have now included a whole new supplementary figure to discuss the distribution of fit initial state updating parameters, and explicitly tested one possible (and popular) generative model of prior updating - the Dynamic Belief model (DBM). First, we found that for about a third of the rat population (47/105), DBM is supported by BIC compared to the elaborated exponential kernel.

For the rest of the rats, we analyzed the fit parameters to infer if they might correspond to a recognizable model of statistical inference. First, we checked if the magnitudes of updating (η s) following wins or losses were significantly different i.e. if post win updating magnitudes tend to be higher/lower compared to post losses. Across the population, we found that the median magnitudes of post-win and post-loss updating showed no significant differences ($p = 0.11$, Wilcoxon signed-rank test), while their signs did differ – post wins tended to induce repeat tendency whereas post losses tended to induce switching. Second, we found that across the population, median timescales of updating (β s) following wins and losses were also not significantly different ($p = 0.46$, Wilcoxon signed-rank test). Then, we sought to analyze the correlation patterns between magnitude of initial state updates following wins and losses - while the DBM population demonstrated a significant correlation ($r = -0.45$, $p = 0.001$), we did not

observe such a pattern in the rest of the population ($r = -0.12$, $p = 0.22$). Further, the timescales of updating following wins and losses showed no significant correlation ($r = 0.02$, $p = 0.78$). However, this was also true for the DBM population ($r = 0.13$, $p = 0.38$).

These observations are potentially consistent with a statistical model in which wins and losses are treated differently during updating (e.g. Hermoso-Mendizabal et al. 2020; Karlsson et al. 2012), owing to the fact that they might carry differential information - while both might contribute to updating current estimates, losses might additionally signal an abrupt change in the environment warranting a “re-drawing” of estimates. While such a model is similar in spirit to the DBM, its exact nature still remains to be fleshed out, and any claims would require further investigation. We also attempted to cluster the parameters to infer if multiple distinct strategies exist within the population, however this did not yield any interpretable results.

The supplementary figure and newly added text to Results section is duplicated below:

Line 266:

To gain further insight into the initial state updating dynamics, we examined the fit parameters controlling the magnitude and timescale of updates (Supp Fig 7). We found that across the population of rats, updates following wins and losses had similar magnitudes, but opposite signs, suggesting a tendency to repeat after wins and switch after losses. We compared these fits to those from a restricted version of the model whose initial state dynamics correspond to optimal updates in a Dynamic Belief Model (Yu, Dayan, et al. 2009, Supp Fig 1) and found that about a third of the population (47/152 rats) were consistent with this form of statistical inference. The remainder of the population did not show a significant correlation between post-win and post-loss parameters, consistent with a statistical model that treats wins and losses differentially (e.g. as in Hermoso-Mendizabal et al. 2020; Karlsson et al. 2012).

Supplementary Figure 7: Interpreting initial state updates through the lens of statistical inference (A) Distribution of initial state updating parameters across the population of rats. Median magnitudes of updating following wins (first two panels) and losses (third panel) were not significantly different in their absolute value (median: $\text{mean}(\eta_{Lw}, \eta_{Rw}) = 0.36$, $\eta_l = -0.35$, $p = 0.11$, Wilcoxon signed-rank test) however they did differ in their signs - wins tended to induce a tendency to repeat (positive X values) while losses induced a tendency to switch (negative X values). Median timescales of updating following wins (0.34) and losses (0.29; fourth, fifth panels) were not significantly different ($p = 0.46$, Wilcoxon signed-rank test). **(B)** Comparison of the ‘‘HISt’’ model with a reduced version that approximates optimal statistical inference in the Dynamic Belief Model (Supp Fig 1, Yu and Cohen 2009). The optimal update rule for the DBM corresponds to a restricted regime of the HISt model, where the magnitude and timescale parameters are set to be the same for all trial types i.e. left-win, right-win, left-loss and right-loss. We compared the fits of this ‘‘Reduced HISt’’ model with the HISt model, and found that it was not supported by BIC in the majority of rats (Left panel: overall bar height denotes the total number of rats for which that model variant scored the lowest BIC score, HISt was winning model in 105/152 rats). Moreover, the unconstrained HISt model performed better on average per trial (Right panel: Model comparison using BIC by pooling per trial BIC score across rats and computing mean. Lower scores indicate better fits. Mean per trial BIC scores across rats were significantly lower for model with HISt, $p = 2.23 \times 10^{-6}$, paired t-test. Error bars are SEM). Overall, this means that while some rats’ (47/152) updating strategy was consistent with optimal inference in a DBM, most rats’ parameters did not occupy this restricted regime. **(C)** Joint distribution of updating magnitude (left) and timescale (right) parameters following wins and losses. Caption continued on next page.

Supplementary Figure 7: (Previous page.) Contrary to the expectation from optimal updating in a DBM, the population of rats did not show a significant correlation between these parameters following wins and losses (magnitude of updating: $r = -0.12, p = 0.22$, timescale: $r = 0.02, p = 0.78$). However, the subset of 47 rats best fit by the Restricted HSt model (not shown) did show a significant correlation in the magnitude parameter alone ($r = -0.45, p = 0.001$). These findings could be potentially consistent with a statistical model in which wins and losses are treated differently for updating (e.g. Hermoso-Mendizabal et al. 2020; Karlsson et al. 2012). This could be due to the differential information signalled by wins and losses - while both are relevant for estimation of current prior, losses might additionally signify an abrupt change in the environment warranting a “re-drawing” of prior estimates. While such a model bears resemblance to DBM, its precise characteristics are yet to be fully defined.

2. The second issue relates to the models of “true lapses”. My intuition is that the deterministic nature of the “inattention variant” would render it very poorly suited to modeling behavioral data, and that the “hybrid variant” is a much stronger and more practical candidate model. However, the hybrid model is fit only to the RT data set, even though as far as I can tell it is just as applicable to the full sample. I would much prefer to see either both models, or just the hybrid model used to analyze the full sample in Supp. Figure 4.

We shared the reviewer’s initial intuition and had indeed tested both the hybrid model and the inattention model on choice data. However, the hybrid model did not significantly enhance the fits to justify introducing an additional parameter compared to the inattention model (as evidenced by the number of

rats with lower BIC for each model variant and the mean BIC score per trial across rats). Furthermore, the inattention model, with the same number of parameters as the HSt model, allowed for a level comparison.

It is also worth noting that the primary finding of this manuscript is the surprising role of HSt in producing lapse rates. We think that Fig 3 convincingly shows the huge variance that is captured by this mechanism . While it is plausible that various versions of true lapse models might better capture the (small) residual

variance, they serve as supplementary findings to this primary result. Considering this, we opted for the more parsimonious variant for our analysis. We hope that the reviewer finds this choice agreeable.

3. Noting that augmenting HSt with the “inattention variant” of true lapses does not change the model fits much, the authors claim on page 14 that “apparent lapses produced by history-dependent initial states (rather than true lapses due to motor error or inattention) are the major driver of history-dependent co-modulations in psychometric thresholds and lapse rates in the dataset.” I do not think this claim is justified by the results they present. If adding a new feature to a model does not change its predictions, this does not mean that we can conclude that this feature is not important. It could also be that the new feature is redundant with previous features of the model. For example, in a linear regression, adding a new covariate that is highly correlated with the previous covariates will not change the model fit, but this does not mean that the new covariate is less important than the previous covariates; instead, it means we cannot distinguish between them.

In this paper, it could be that choice-history effects on the accumulation process are indistinguishable from choice-history effects on inattention-driven lapses. To demonstrate that this is not the case, I would suggest fitting a model where choice history can only affect lapses. Under the author's hypothesis, this model should be much worse than any model that allows choice history to impact the accumulation process. Again, I would prefer to see this comparison done with the hybrid lapse model as well as (or just instead of) the inattention variant.

We thank the reviewer for this comment, as it has helped us clarify our intended message. Indeed, we agree that in order to show that H1St is the primary contributor to joint history-dependence of lapses and thresholds, one needs to show both sufficiency (i.e. H1St alone captures the modulation compared to H1St + inattentive lapses) and necessity (removing H1St and only having inattentive lapses that are history-dependent performs much worse). We have now added this model, which performs much worse, and this has let us make a clearer statement about the contribution of H1St compared to inattention. The newly added supplementary figure is replicated below:

Supplementary Figure 6: An accumulator model with history modulation of just true lapses performs much worse. To demonstrate that the trial-history effects on the initial state of the accumulator are distinguishable from trial-history effects on true lapses, we compared the HlSt models with a model variant in which trial-history exclusively modulated lapses. This is important since the BIC scores were similar for model variants with history modulation of true lapses (HlSt + Inattention, Supp Fig. 4). We found that this model performs much worse as measured by BIC (A), AIC (B) and Out-of-sample loglikelihood (C). These plots show the distribution of rats that are best fit by these model variants based on these individual metrics. (D) Difference in BIC scores from the model variant with best score are shown for each individual rat. These results provide strong evidence that history modulation of the initial state plays a crucial role in achieving better model fits and effectively capturing the comodulation of threshold and lapse rate parameters. Conversely, the true lapse modulations make only a minor contribution to the overall performance.

4. Finally, it would be interesting to know what proportion of lapses are “apparent” versus “true” in the rodents’ choices. The “true” lapse rate is encoded directly in the parameter κ , while the “apparent” lapse rate could presumably be extracted from the implied psychometric curves of the accumulator model.

Thank you for this suggestion. We compared the true lapse rate (κ) fit by the model with the total lapse rate inferred by fitting model-produced choices with (logistic) psychometric curves. We found that true lapses only had a median contribution of 32% to the total lapse rate, albeit with a wide distribution across rats.

Minor issues and clarifications:

- Are the parameters for each rat assumed fixed across sessions? It seems so but this should be explicitly noted in the methods section.

We have explicitly noted this in the Methods, under the section on Model Fitting:

Line 582:

Throughout this manuscript, we assumed that for each rat, the parameters remain fixed across all sessions. So one set of parameters were fit to each rat for each model variant.

- Pg 10, 188, “Across rats, 17% of total lapse rates are modulated by trial history”, and Supp. Fig 3D: I have no idea what analysis was done here. How many lapse rates are calculated for each rat, and how? Apologies if I am missing this through my own inattention.

Apologies for the confusing wording - one set of lapse rate was calculated for each rat (as detailed in Methods/Psychometric curves). In this analysis, we are reporting the fraction of this variance which is history modulated (Supp Fig 2B, Methods/Psychometric curves/History modulation of psychometric parameters). We have attempted to clarify this the in the text:

Line 197

Across rats, on average 17% of lapses are modulated by trial history and therefore could potentially reflect apparent rather than true lapses (Supp Fig.3D).

Supp fig 3D captions (Page 43):

Histogram of history-modulated lapse rates as a fraction of total lapse rates. A sizeable portion of the population had non-zero fractions, suggesting that history-dependence could potentially account for substantial lapse rate variance.

- Supp Fig 2A, “Blue and pink curves are conditioned on past left/right wins”. All past left/right wins, or only the last trial being a left/right win? In behavior plots, it seems to be conditioned on the last trial, but here it is ambiguous. Also, please give explicit parameter values instead (or in addition to) high/low, etc.

We have clarified that it is only conditioned on the past trial being a left/right win, and added the parameter values.

- Fig 3D and Supp Fig 4E are much less informative than they could be. I recommend showing the distribution of BIC differences across rats. Apply this to any other max BIC plots I have forgotten.

Thanks, we have included plots with distributions of BIC differences across rats in the supplement (Supplementary Figure 6 - also replicated above). This figure contains all the different model variants are comparisons included in the manuscript for the choice dataset.

- Page 13, 227-230, “Across individuals, the model with H1St captured...”: I do not understand what analysis was actually done here. It must refer to the results displayed in Fig 3F & G, but this is not stated in the text. The description of the analysis in the figure caption is more clear, but still needs elaboration. How are history modulations of thresholds and lapse rates quantified from the H1St model?

Thanks for raising this, we have now added reference to the figure, relevant methods and a short description of what's being done.

Line 236

Next, we examined the extent to which these modulations were captured across individual rats (Fig 3F,G). We quantified these history modulations as follows: “threshold modulations” are defined as the horizontal distance between the midpoints of psychometric curves conditioned on previous wins and losses, and “lapse rate modulation” as the vertical distance between the asymptotes of these curves (Methods: History modulation of psychometric parameters, also see Supp Fig 2B). This was done separately for model-predicted and rat choices and then compared.

- Early in the methods (e.g., bottom of p21), x is used as the accumulated evidence variable and ϵ for the increments. However, starting on p25, the authors use the notation of Brunton et al., where accumulated evidence is $a(t)$, clicks are δ , and the definition of η is different than the one used in the filtering model for trial history effects. It would be helpful if the authors harmonize their notation here. We thank the reviewer for pointing this out, we have now harmonized the notation in the manuscript.

- The joint model of choices and RTs (page 27-28) contains insufficient detail on the fitting process. The integral in the penultimate equation presumably does not have a closed form solution, so how was the model estimated?

Thanks for this comment, the likelihood was computed numerically using previously established methods. We have modified text in the Methods section to clarify this:

Line 597:

We followed previously established methods to compute the probability distribution of $x(t)$ for computing the likelihood (Brunton et al., 2013; DePasquale et al., 2021). This involves expressing the temporal dynamics of the probability distribution as a Fokker-Planck equation and then computing the solution numerically, by dividing $P(x(t))$ into a set of n discrete spatial bins and determining how mass moves after a discrete temporal interval Δt . The transition matrix for the discrete time dynamics and a full description of the methods can be found in these studies.

Reviewer #4 (Remarks to the Author):

This study aims to provide a more unified view of certain common suboptimalities in decision-making behaviors, typically described as biases and lapses, in terms of mistaken beliefs about the stationarity of the task environment that drive history-dependent adjustments in behavior. They develop a computational model based on these ideas and show that it does a nice job fitting rodent data.

I think the basic ideas are reasonable, interesting, and valuable to the field (that can always use more precise and accurate models of behavior); the model is appropriate and provides a clear, quantitative description of the data; and the paper is well written. All that said, I also have several major concerns, detailed below, that should be addressed:

We are thankful to the reviewer for these kind words, a point-by-point response follows.

1. Much of the framing makes the basic ideas sound more novel than I think they are. I kept wanting a simpler description of the phenomenon and the model, something along the lines of, “we’ve known since at least Laming (1968) that decision-makers often show suboptimal sequential biases, and we’ve known from countless studies that biases can have several well-known effects on psychometric (and chronometric) functions, so if you put those together and the sequential biases are big enough you can get these effects.” Again, I think putting it together is useful, but from a theoretical perspective the ideas aren’t new, and I think better framing them in terms of what is known will make them more accessible. Some related points:

We agree with the reviewer that descriptions and models of suboptimal sequential biases have a rich history in the literature; we have redoubled efforts to emphasize this by expanding our Discussion, and citing more references in Introduction and Results.

We have added a new paragraph to the Discussion elucidating recent work that comes close to ours, by examining initial point variability in the large bias regime (Kilpatrick) or heavy tailed psychometrics that result from variable precision across trials (Shen & Ma). However, to our knowledge, we are *not* aware of work that directly examines the effects of such sequential biases on lapse rates, or links them to the separate literature in causes of lapses - this is where our paper’s novel contribution lies - if the reviewer has references to such work we would be happy to cite it.

Line 382:

A number of previous studies have hinted at the performance-limiting effect of sequential biases, variability in initial points and/or sensitivity across trials (Bogacz 2006, Shen & Ma 2019, Nguyen 2019). Nguyen et al 2019 examined the optimal decision making strategy under a non-stationary generative model, and arrived at psychometric curves similar to the heavy-tailed curves produced by our model. Similarly, Shen & Ma 2019 examined decision-making under variable “precision” across trials, which also yields heavy-tailed psychometrics, trading off against lapse parameters. However, to our knowledge, ours is the first study to directly examine the effect of sequential biases on lapse rates, and link

the two relatively separate literatures. Our model formulation shares some features with previous work on sequential biases, albeit with some distinct features - our model is a Drift Diffusion Model with history-dependent initial states (similar to Nguyen et al 2019, but unlike Kim et al 2017, who use an adaptive LATER model) adapted to discrete stimuli for the purpose of trial-by-trial modeling. Our model's initial states are a continuous variable, unlike Urai et al 2019, whose initial states take on one of two possible discrete values. Also, our model's initial states are set by a flexible exponential filter on several past choices and outcomes, unlike Nguyen et al 2019, Kim et al 2017, Yu et al 2008 and other variants of the Dynamic Belief Model, albeit reducing to them for certain restricted parameter regimes.

We have also included additional references in our introduction and discussion sections where we outline past theoretical work on such biases. The lines are reproduced here:

From Introduction (Line 28):

History biases may arise due to a strategy that is optimized for naturalistic settings, where continual learning of priors, action-values, or other decision variables helps agents adapt to changing environments, but is maladaptive in experimental settings where the statistics of the environment are stationary (Yu and Cohen, 2009; Molano-Mazon et al., 2021). To date, decision-theoretic models have accommodated history biases by modeling them as a biasing factor on the perceptual evidence that drives choices (Laming, 1968; Ratcliff and Rouder, 1998; Bogacz et al., 2006; Busse et al., 2011; Goldfarb et al., 2012; Kim et al., 2017; Gardner, 2019; Urai et al., 2019; Hermoso-Mendizabal et al., 2020).

From Discussion (Line 399):

In our treatment, we only considered history-dependent updates to the initial state of a DDM. Such a mechanism is normative under non-stationary beliefs about the prior, which is our favored interpretation as it aligns with other studies of history biases (Gold, Law, et al., 2008; Yu and Cohen, 2009; Summerfield and Koechlin, 2010; Goldfarb et al., 2012; Mulder et al., 2012; Abrahamyan et al., 2016; Kim et al., 2017; Molano-Mazon et al., 2021). Nevertheless, these updates may also reflect other heuristic strategies (Gigerenzer and Gaissmaier, 2011) which we accommodate using our flexible parameterization of initial state updates. Animals may entertain non-stationary beliefs about other elements of the decision process, such as the rewards or likelihoods (Dayan and Daw, 2008; Mendonça et al., 2020; Lak et al., 2020; Pisupati et al., 2021).

a. The technically most interesting/relevant part of the story is exactly how a (true or apparent) lapse is measured. As the authors note, a lapse is considered an “asymptotic” error – but exactly how the asymptote is defined and measured can vary widely across studies. Of course, apparently small differences in where/how the asymptote is measured can have a major role in how susceptible the measurement is to the kind of biasing influence they posit (e.g., consider an initial bias of something like 0.731, corresponding to log-odds of ~1; if the “asymptote” is measured for “strong” evidence corresponding to log-odds of 3, then the bias could provide a very measurable change from ~95% choices to ~88% choices; however, if stronger evidence corresponding to a log-odds of ~6 is used, then the effects of the biasing term might be negligible, going from 99.8% to 99.3% choices). It might be

useful to provide some insights into how these considerations relate to the conditions in which one might or might not expect to see the effects reported.

b. Relatedly, it would be useful to provide more intuition for why/when the model gives predictions that seem to correspond to either a flattened asymptote on the psychometric curve that traditionally would be thought of as a “lapse” (e.g., Fig. 3B, right panel) or curves that do not seem to have yet reached an asymptote and thus would not traditionally be thought of as being useful for providing a good estimate of lapses (e.g., Fig. 3C, right panel).

- (a) We thank the reviewer for this suggestion, and wholeheartedly agree that the method/stimulus strengths used to measure asymptotes play an important role in their estimation. To make this important point evident, we have added a new supplementary figure showing the extent to which logistic fits to the psychometric curve accurately recover true lapse rates for (i) different number of trials (ii) maximum evidence strength considered. We compare these results to when asymptotes are estimated by measuring performance at the extreme stimulus strength alone and find this to be far less reliable. We make comparisons between these methods, both for when the psychometric curve is unbiased and biased. We find that the method of fitting logistic functions to the psychometric curves for measuring asymptotes is reliable, especially for regimes in which our data resides. In our manuscript, we have exclusively relied on this method for measuring lapse rates. The newly added supplementary figure is duplicated below.
- (b) Thanks for this suggestion. We have added new text to provide more intuition for when the model produces heavy-tailed psychometric curves, as well as the conditions under which these curves would be estimated as lapse rates versus not.

Pg 42 (Supp Fig 2 captions):

It is important to note that, apart from the final case, the apparent lapses stem from the fact that diverse initial states lead to psychometric curves characterized by varying “sensitivities”, resulting in psychometric curves with heavy tails. When these curves are approximated using the logistic function, the heavy tails are accounted for as lapse rates. If measurements were conducted for stimulus strengths that are even higher, the inherent heaviness of the psychometric function’s tails would become evident, rendering logistic functions inadequate for providing a suitable fit. Measures of overdispersion (e.g. Schütt et al. 2016) can be used to detect such heavy-tailedness, when measuring choices at higher and higher stimulus strengths is not feasible.

Line 130 (Results):

Importantly, these “apparent lapses” contribute to lapse rates when heavy-tailed psychometric curves are approximated by a logistic function. However, this approximation is bound to be inadequate if measurements were made for even higher stimulus strengths, making the heaviness of the tails even more evident.

Supplementary Figure 4: **Logistic fits to psychometric curves reliably recover performance asymptotes i.e lapse rates** For this study, it is crucial to reliably estimate the asymptotes of psychometric curves in order to measure lapses. We confirmed that the logistic fits (see Methods, Psychometric curves) consistently yield reliable recovery of lapse rates across various conditions. These conditions include diverse total trial counts (depicted in middle panels A and B) and variations in the maximum stimulus strength at which choice behavior is measured (illustrated in bottom panels A and B). We find that the method accurately estimates the true lapse rates in scenarios when choices are biased (A) as well as unbiased (B). We contrast the performance of this method with an alternative approach of estimating asymptotes by simply measuring the performance at easiest stimuli (i.e. maximum stimulus strengths) e.g. as in Wang, Montanède, et al. 2019. This approach yields biased estimates particularly when the trial counts are low or when choices are measured only at low stimulus strengths.

Across all simulations, the sensitivity of psychometric curves was set to 0.125, similar to values typically seen in our dataset (Fig 2C). For simulations involving varying trial counts (middle panels), the maximum stimulus strength was set at 40. Simulations exploring distinct maximum stimulus strengths (bottom panels) were carried out with 10,000 simulated choices. The top panels represent choices simulated with true lapse probability of 0.2. Each point corresponds to estimates derived from individual simulated datasets, error bars have been omitted for clarity of visualization.

c. It also seems like it could be useful to be clearer in places about the definition of “lapses.” For example, the Discussion states that “Lapse rates are often considered to be a mixed bag comprising several different noise processes, yet most studies so far have focused on one or more of these component processes in isolation.” This seems vague to me and misses the key point that the traditional definition of lapses emphasizes that they are based on factors that are entirely separate from the decision process (e.g., “finger slips”).

We thank the reviewer for pointing this out. We completely agree with the reviewer's definition of lapses - as being evidence-independent errors caused by processes that are separate from the decision process - indeed this is the one that we rely on in our introduction. We have clarified this definition in our box and discussion, in an effort to keep definitions consistent across the manuscript.

Introduction (Line 38):

A second widely-recognized but less studied suboptimality is the tendency to “lapse”, or make (asymptotic) errors that are immune to strong evidence (refs...). Because lapses appear to be evidence-independent, they are assumed to arise from nuisance mechanisms that are separate from the perceptual decision-making process and are often imputed to ad-hoc noise sources such as inattention, motor errors etc.

Box (page 8):

True lapse: A true lapse is a stochastic, evidence-independent choice that arises from cognitive processes **entirely separate from the decision process, such as inattention or motor error.**

Discussion (Line 427):

Lapse rates are often considered to be a mixed bag comprising several different noise processes **separate from the decision process**, yet most studies so far have focused on one or more of these component processes in isolation.

d. As noted above, the main behavioral effect driving the results seems to be that the rats have pretty big, suboptimal sequential biases. Seems like it would be useful to discuss that finding (and the degree to which it likely does and does not support generalization of the findings to other species/tasks/conditions) in more detail.

Thanks for raising this point, we have amended the discussion to include the different settings in which we expect our results to generalize and where we expect other processes to be at play.

Line 416:

Another crucial possibility, trial-to-trial variability in drift rates - is known to give rise to longer error RTs than correct RTs (Ditterich, 2006a; Ditterich, 2006b; Drugowitsch, Moreno-Bote, et al., 2012; Nguyen and Reinagel, 2020), which is a signature often reported in monkeys and humans (Roitman and Shadlen, 2002; Shevinsky and Reinagel, 2019). We did not observe the reaction time signatures of drift rate variability in our dataset, instead we identified signatures of initial state variability, where error RTs were shorter than correct RTs, rather than longer. However, drift rate updates may represent an alternative mechanism through which history-modulated apparent lapses could occur in other datasets. It is worth noting that certain task designs include efforts to actively measure and counter trial history biases. In such cases, lapses may still occur, likely due to exploration or inattention. In this manuscript, we refer to lapses caused by these factors “true lapses”, as they cannot be explained by fluctuations in DDM-related parameters.

(Relevant bits from Discussion already part of manuscript) Line 359:

History biases in perceptual decision making tasks have been modeled using initial state updates to DDMs in humans and non-human primates (Gold, Law, et al., 2008; Goldfarb et al., 2012; Zhang et al., 2014). These studies tended to have relatively small magnitudes of history bias, and miniscule lapse rates, hence being well captured by small deviations in the initial state of a DDM, which largely yield horizontal shifts in the psychometric function. This regime of initial state updates is well approximated by a logistic function with additive biases, which is the dominant descriptive model used to characterize history-dependent psychometric curves (Busse et al., 2011; Carandini and Churchland, 2013; Fründ et al., 2014; Abrahamyan et al., 2016; Gardner, 2019; Pinto et al., 2018; Odoemene et al., 2018; Urai et al., 2019; Hermoso-Mendizabal et al., 2020; Roy et al., 2021; Ashwood et al., 2022; Bolkan et al., 2022). However, as we demonstrate, when deviations in the initial state are large, this logistic approximation breaks down. This fact has been overlooked in much of the literature. Consequently, even in datasets with large history biases and lapses, the logistic formulation continues to be favored (Odoemene et al., 2018; The International Brain Laboratory et al., 2021; Roy et al., 2021; Ashwood et al., 2022), albeit requiring additional components. **Such effects tend to be prevalent in rodents but not human or non-human primate behavior.** Our demonstration predicts that the full range of initial state effects should resemble concurrent, trial-by-trial changes in both threshold and sensitivity parameters of the logistic function. Indeed, Ashwood et al. 2022 found that apparent lapses in several rodent datasets can be better captured by runs of trials with such concurrent modulations, yielding biased “disengaged” states.

2. It also would be nice to provide some more background about what is and is not new about the basic model. For example, Kim et al (J Neurosci, 2017) also used an “accumulation to-bound model with history-dependent updates to the initial state of the accumulator” where the updates were based on “Bayesian learning of priors under the belief that the prior probabilities of the two hypotheses can undergo unsigned jumps.”

We agree, it is important to have clarity on the aspects of our model that differ from previous models of sequential biases. We have added a line to the discussion detailing these (also copied in a response above):

Line 390

Our model formulation shares some features with previous work on sequential biases, albeit with some distinct features - our model is a Drift Diffusion Model with history-dependent initial states (similar to Nguyen et al 2019, but unlike Kim et al 2017, who use an adaptive LATER model) adapted to discrete stimuli for the purpose of trial-by-trial modeling. Our model’s initial states are a continuous variable, unlike Urai et al 2019, whose initial states take on one of two possible discrete values. Also, our model’s initial states are set by a flexible exponential filter on several past choices and outcomes, unlike Nguyen et al 2019, Kim et al 2017, Yu et al 2008 and other variants of the Dynamic Belief Model, albeit reducing to them for certain restricted parameter regimes.

Minor comments:

1. I found the description of Eq. 3 (Lines 110–121) to be a bit confusing, in several ways: a) saying “small deviations in the initial state largely manifest as additive biases to the evidence, shifting psychometric

curves horizontally towards the option favored by the initial state” seems like an odd way of describing an effect that is most definitely not an additive bias to the momentary evidence (which can of course be modeled explicitly) but rather a change in total evidence (maybe instead of “largely manifest as” you can say “resemble” if you really want to make this point); and b) calling a shift in this kind of psychometric function (where choice is on the y axis) as affecting “threshold” also may be a bit confusing, since “threshold” is usually a measure of sensitivity, which as described a few lines below is reflected in the slope, not the shift, of these curves.

- a) Agreed - in order to clarify this point, we have changed this sentence to:
“small deviations in the initial state largely **resemble** additive biases to the **total** evidence”
- b) Agreed - in order to avoid confusion, we have added a note about our use of the word “threshold” to denote the x-axis value at the inflection point of the psychometric, and “slope” to denote the sensitivity or slope of the curve at this inflection point, following the convention from Wichmann & Hill (2001). We have added this as a note in the main text (line 121) as well as in Methods.

Under Methods/Psychometric Curves:

Throughout this manuscript, we follow the convention from Wichmann and Hill (2001) and use “threshold” to denote the x-axis value at the inflection point of the psychometric curve, and “slope” to denote the sensitivity or slope of the curve at this inflection point.

REVIEWERS' COMMENTS

Reviewer #1 (Remarks to the Author):

The authors have properly addressed all my comments.

Reviewer #2 (Remarks to the Author):

I have read the authors responses to the reviewer's comments and the revised version of the manuscript. The authors did an excellent job in addressing my questions. I have no further comments.

Reviewer #3 (Remarks to the Author):

I appreciate the authors' thoughtful and thorough responses to my questions. I have no further concerns and support publication of the results.

Reviewer #4 (Remarks to the Author):

The authors have done a very nice job of responding to all of the reviewers' concerns. I commend them for an interesting and useful study.

REVIEWERS' COMMENTS

Reviewer #1 (Remarks to the Author):

The authors have properly addressed all my comments.

Reviewer #2 (Remarks to the Author):

I have read the authors responses to the reviewer's comments and the revised version of the manuscript. The authors did an excellent job in addressing my questions. I have no further comments.

Reviewer #3 (Remarks to the Author):

I appreciate the authors' thoughtful and thorough responses to my questions. I have no further concerns and support publication of the results.

Reviewer #4 (Remarks to the Author):

The authors have done a very nice job of responding to all of the reviewers' concerns. I commend them for an interesting and useful study.

We thank all the reviewers for their kind words and for thoughtfully engaging with this work. We think that their comments and suggestions have greatly helped strengthen the manuscript.